# All-in-one porous membrane enables full protection in guided bone regeneration

Shuyi Wu [1,4], Shulu Luo[1,4], Zongheng Cen[2,4], Qianqian Li[1], Luwei Li[1], Weiran Li[1], Zhike Huang[3], Wenyi He[2], Guobin Liang[1], Dingcai Wu [2] ✉, Minghong Zhou[3] ✉ & Yan Li [1] ✉

The sophisticated hierarchical structure that precisely combines contradictory mechanical and biological characteristics is ideal for biomaterials, but it is challenging to achieve. Herein, we engineer a spatiotemporally hierarchical guided bone regeneration (GBR) membrane by rational bilayer integration of densely porous N-halamine functionalized bacterial cellulose nanonetwork facing the gingiva and loosely porous chitosan-hydroxyapatite composite micronetwork facing the alveolar bone. Our GBR membrane asymmetrically combine stiffness and flexibility, ingrowth barrier and ingrowth guiding, as well as anti-bacteria and cell-activation. The dense layer has a mechanically matched space maintenance capacity toward gingiva, continuously blocks fibroblasts, and prevents bacterial invasion with multiple mechanisms including release-killing, contact-killing, anti-adhesion, and nanopore-blocking; the loose layer is ultra-soft to conformally cover bone surfaces and defect cavity edges, enables ingrowth of osteogenesis-associated cells, and creates a favorable osteogenic microenvironment. As a result, our all-in-one porous membrane possesses full protective abilities in GBR.

Precise coupling of different or even contradictory material properties and biological characteristics, such as stiffness and flexibility, ingrowth barrier and ingrowth guiding, as well as antibacteria and cell-activation, is of great demand in the field of tissue engineering but remains challenging[1–4]. As the pivotal material for the guided bone regeneration (GBR) technique to augment bone, GBR membranes applied between the gingiva and alveolar bone in the oral cavity can be considered as an ideal model for studying the biological applications of materials implanted at the soft-hard tissue interface[5]. Due to the opposite properties of the adjacent interfaces, GBR membranes must meet stringent criteria in materials design. The desired GBR membrane demands a sophisticated integration of properties: sufficient rigidity to maintain the space for tissue regeneration, but good softness for tissue adaption; effective resistance to the inward preoccupied soft tissue

cells, but good capability of guiding the ingrowth of osteogenesis-associated cells; defense against bacteria in a microbial-rich intraoral environment, but promotion towards osteogenesis in the bone defect[6,7].

Nowadays, the most widely used GBR membranes in clinical practice are collagen membranes, such as Geistlich Bio-Gide® [8], Zimmer CopiOs® [9], and ZH-Bio Heal-All® [10], which rely on the porous skeleton formed by stacking of collagen fibers to block soft tissue cells[11]. Due to the variability of clinical cases and the microbial richness of the oral environment, the shortcomings of collagen membranes are increasingly magnified[12]. Collagen could be swollen and softened by saliva and blood, resulting in limited space maintenance in practical applications[13,14]. Moreover, infection is one of the main causes of GBR failure, particularly in patients with periodontitis, maxillofacial defects,

[1]Hospital of Stomatology, Guanghua School of Stomatology, Guangdong Provincial Key Laboratory of Stomatology, Sun Yat-sen University, 510055 Guangzhou, P. R. China. [2]Key Laboratory for Polymeric Composite and Functional Materials of Ministry of Education, School of Chemistry, Sun Yat-sen University, 510006 Guangzhou, P.R. China. [3]Medical Research Institute, Guangdong Provincial People's Hospital (Guangdong Academy of Medical Sciences), Southern Medical University, 510080 Guangzhou, P. R. China. [4]These authors contributed equally: Shuyi Wu, Shulu Luo, Zongheng Cen. ✉e-mail: wudc@mail.sysu.edu.cn; zhouminghong@gdph.org.cn; liy8@mail.sysu.edu.cn

and other poor microecological environment, as well as special patients with low immunity (e.g., diabetes). However, the commonly used collagen membranes lack components or structures that resist bacterial invasion[11], let alone the quickly weakened barrier function caused by the rapid degradation of collagen[15-17]. Therefore, the existing GBR membranes are still far from the ideal soft-hard tissue interface biomaterial (Fig. 1a).

Efforts have been made to address the aforementioned issues, but most efforts are largely limited to addressing individual issues rather than considering them comprehensively. For example, titanium mesh is used to improve space maintenance. However, its excessive rigidity can easily lead to gingiva dehiscence and hinder its proper attachment to the bone surface to form a seal[18]. Some GBR membranes have been endowed with antibacterial properties, but they neglect the complexity and spatiotemporal characteristics of bacterial infection and then fail to effectively resist bacterial invasion at multiple stages (including adhesion, colonization, and penetration) and in multiple directions[19-21]. Therefore, how to delicately couple the seemingly opposite mechanical, structural, and biological properties in a GBR membrane to provide full protection during bone regeneration is a vital scientific and clinical question.

Herein, based on the multiple adaptations including mechanics, pore structure, and antibacterial performance, we have created a spatiotemporally hierarchical GBR membrane, which can comprehensively solve the clinical issues throughout GBR processes (Fig. 1b). The dense layer of the membrane facing the gingiva shows Young's modulus comparable to that of the gingiva, so it has a sufficient space maintenance capacity, and would not damage the gingival tissue or cause a stress shielding effect on the osteogenic area. The loose layer facing the alveolar bone is ultra-soft to conform to various morphologies of bone surfaces and seal the edge of the defect cavity. With the above asymmetric design, we integrate rigidity with flexibility to meet the desirable mechanical properties of the GBR membrane. In terms of the hierarchical porous structure, the small pores of the dense layer can continuously barrier both fibroblasts and bacteria, while the largely porous framework of the loose layer is suitable for the ingrowth of osteogenesis-associated cells. More importantly, through the asymmetric antibacterial function, the dense layer acts as a powerful defense with multiple mechanisms, including release-killing, contact-killing and anti-adhesion, to prevent bacterial invasion from the outside of the membrane; the loose layer moderately prevents bacteria leakage from other directions and creates a favorable osteogenic microenvironment. With a well-orchestrated combination of the hierarchical porous structure and multi-dimensional antibacterial function, the quintuple protections against bacteria are complementary and indispensable to resist bacterial invasion strongly and comprehensively. As a result, our all-in-one GBR membrane enables full protection and osteogenic promotion, making it a promising GBR material in clinical bone augmentation.

## Results

### Fabrication of asymmetric porous bilayer membrane

The preparation process of our asymmetric porous bilayer membrane includes the construction of dense layer and loose layer (Fig. 2a). We elaborately select bacterial cellulose (BC) membrane with a 3D porous nanonetwork structure as the substrate. Hairy polyacrylamide (PAM) chains are grafted on BC nanofibrils of BC membrane to construct PAM-grafted BC membrane (BC-*g*-PAM), and further chlorinated with sodium hypochlorite (NaClO) to obtain N-halamine functionalized BC dense layer (BC-*g*-PNCl). A viscous aqueous solution of chitosan (CS) and hydroxyapatite (HAP) is then drop-coated uniformly on the surface of BC-*g*-PNCl layer and lyophilized to form the loose layer (CS-HAP), mimicking the synergistic combination of organic and inorganic components naturally present in the bone[22,23].

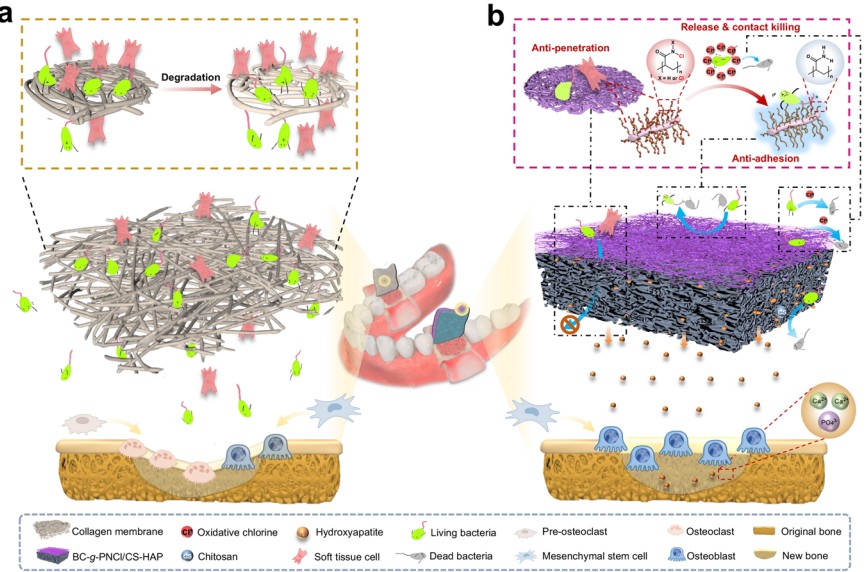

**Fig. 1 | Schematic illustration of structures and protective behaviors in commercial collagen membrane and BC-*g*-PNCl/CS-HAP. a** Commercial collagen membrane usually lacks efficient structures and components to resist bacterial invasion, presents a high affinity for bacteria and possesses large pores, so it cannot effectively block bacterial invasion and has poor cell barrier ability, giving rise to poor osteogenic effect in the complex oral environment. The above situations will further deteriorate after the rapid degradation of the collagen membrane. **b** With well-integrated contradictory structures and properties, BC-*g*-PNCl/CS-HAP can provide full protection for bone regeneration in oral environment. For example, the dense layer is designed to possess multiple defenses, including the long-lasting release-killing and contact-killing abilities of N-halamine, the combined anti-adhesion function of BC backbones and in situ dechlorination-generated amide groups, and the stable anti-penetration capacity of the nanoscale porous skeleton to mainly prevent the invasion of bacteria from the outside to the inside of the membrane. The loose layer is composed of mildly antibacterial CS, further preventing bacteria from the membrane edges and the infected areas inside the alveolar bone. Moreover, the small pores of the dense layer can effectively barrier fibroblasts, while the largely porous skeleton of the loose layer can provide a good scaffold for bone regeneration. There simultaneously exists the good space maintenance capacity of BC-*g*-PNCl and the osteogenic promotion ability of CS-HAP. Thus, depending on full protection and osteogenic promotion, our all-in-one GBR membrane can well balance multiple contradictions to achieve bone regeneration.

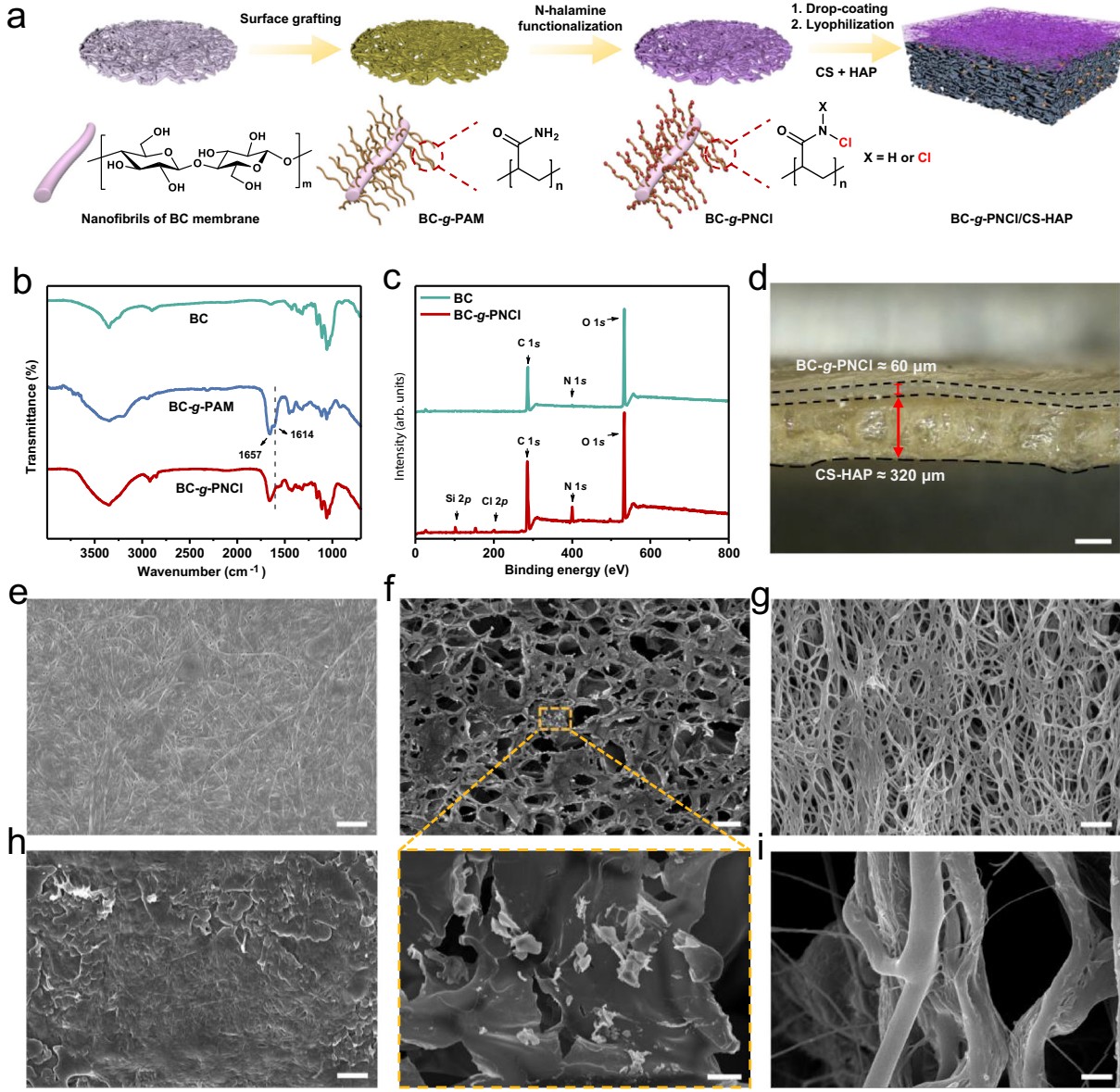

**Fig. 2 | Synthesis procedure and structure characterization. a** BC-*g*-PNCl dense layer towards the soft tissue was prepared by grafting of PAM on nanofibrils of BC membrane, followed by N-Cl functionalization of PAM chains. CS and HAP were introduced into the surface of BC-*g*-PNCl dense layer through drop-coating and lyophilization to form CS-HAP loose layer facing the bone defect. **b**, **c** FTIR spectra of BC, BC-*g*-PAM, and BC-*g*-PNCl (**b**) and XPS spectra of BC and BC-*g*-PNCl (**c**), proving that the N-halamine functionalized BC-*g*-PNCl layer is successfully constructed. **d** Side view macroscopic image showing asymmetric bilayer structure and thickness of BC-*g*-PNCl/CS-HAP (scale bar = 200 μm). **e**–**g** Top view SEM images of BC-*g*-PNCl layer (scale bar = 1 μm) (**e**), CS-HAP layer of BC-*g*-PNCl/CS-HAP (upper scale bar = 50 μm, lower scale bar = 5 μm) (**f**) and dense layer of the commercial CM (scale bar = 1 μm) (**g**). **h**, **i** SEM images of BC-*g*-PNCl layer of BC-*g*-PNCl/CS-HAP (**h**) and dense layer of CM (**i**) after 4 weeks of simulated clinical immersion (scale bars = 1 μm). SEM observation was repeated three times independently, yielding similar results.

## Structural and mechanical characteristics

Fourier transform infrared (FTIR) spectra (Fig. 2b) show absorption peaks of amide groups at 1657 and 1614 cm$^{-1}$ in BC-*g*-PAM, indicating that PAM is successfully grafted[24,25]. The peak at 1614 cm$^{-1}$ is weakened in BC-*g*-PNCl, indicative of the success of N-halamine functionalization[26]. Furthermore, the spectra of X-ray photoelectron spectroscopy (XPS) (Fig. 2c) exhibit a new peak of Cl 2*p* at 200.5 eV in BC-*g*-PNCl compared to BC, confirming the presence of N-halamine groups[27,28]. As shown in elemental mapping in Supplementary Fig. 1, the chlorine elements from N-halamine polymers are homogenously distributed in BC-*g*-PNCl layer, while the phosphate, calcium, carbon and nitrogen elements from HAP and CS are evenly distributed on CS-HAP layer. Based on the micromechanical interlock resulting from the penetration of CS-HAP solution into the dense layer network, the

targeted product BC-*g*-PNCl/CS-HAP has a tightly bound bilayer structure (Fig. 2d), which can be maintained even under high-speed vibration (Supplementary Movie 1). The thickness of BC-*g*-PNCl and CS-HAP layers is about 60 and 320 μm, respectively, making the overall thickness of BC-*g*-PNCl/CS-HAP similar to a commercial Bio-Gide® collagen membrane (CM, normally 300–500 μm)[29,30].

Scanning electron microscopy (SEM) images clearly display a hierarchical porous structure of BC-*g*-PNCl/CS-HAP (Fig. 2e, f). BC-*g*-PNCl layer exhibits a 3D porous nanonetwork structure with an average pore size of about 79 nm (Supplementary Fig. 2a), which is similar to the pore size of BC (Supplementary Fig. 3) but significantly different from the relatively large pore size of the dense surface of commercial CM (Fig. 2g). This well-designed pore structure provides BC-*g*-PNCl dense layer with excellent resistance to the invasions of soft tissue cells

(generally >10 μm in diameter[31]) and even bacteria (mostly >0.2 μm in diameter[32], e.g., 0.5–1.0 μm for *Staphylococcus aureus* (*S. aureus*) in diameter[33]) without impairing the exchange of oxygen and nutrient. Furthermore, substrate surfaces with much smaller pores than microbial cells can reduce the contact area between bacteria and substrate surfaces, induce bacteria-surface mutual repulsion, and then weaken biofilm formation[34–36]. After 4 weeks of in vitro simulated clinical immersion, the pore size of BC-*g*-PNCl shows no significant change (Fig. 2h), enabling durable barrier properties during bone regeneration. However, the pore size of CM dense surface is significantly enlarged (Fig. 2i) because the fibers degrade, fuse, and/or conglutinate rapidly. On the contrary, CS-HAP loose layer that has a 3D interconnected porous micronetwork structure (Fig. 2f) with an average pore size of about 49 μm (Supplementary Fig. 2b) is favorable for osteoblast infiltration and integration[37]. BC-*g*-PNCl dense layer has obviously lower surface roughness than CM (Fig. 3a–c), which helps to reduce the initial bacterial adhesion area and enhance the self-cleaning ability[38,39]. In contrast, the surface roughness of CS-HAP loose layer is higher than that of CM (Fig. 3d–f), providing a more favorable surface for osteoblast attachment[40].

BC-*g*-PNCl serves as the mechanical support of BC-*g*-PNCl/CS-HAP and its Young's modulus is 57.3 MPa in a wet state (Fig. 3g), which is much more compatible with human gingiva (20.0–54.7 MPa[41]) than CM (13.3 MPa). When the same load (1 g) is applied, the wet BC-*g*-PNCl/CS-HAP only slightly deforms and quickly returns to its original form after removing the loading (Fig. 3h and Supplementary Movie 2), whereas the wet CM collapses completely and irreversibly (Fig. 3i and Supplementary Movie 3). The wet BC-*g*-PNCl/CS-HAP can even withstand

loads of over 2.5 g (71 times its own dry weight), while the wet CM can only withstand loads below 50 mg (6 times its own dry weight). Thus, BC-*g*-PNCl/CS-HAP is strong enough to withstand external pressure to maintain the osteogenic space. Moreover, BC-*g*-PNCl is not as rigid as titanium mesh (normally at GPa level), avoiding soft tissue dehiscence and stress shielding on bone[42,43]. Young's modulus of the wet CS-HAP is $3.3 \times 10^{-3}$ MPa, which possesses a similar advantage to CM in conformally covering the bone surface adjacent to the defect cavity to seal the defect area[6]. The wet BC-*g*-PNCl/CS-HAP membrane exhibits higher tensile strength and smaller elongation at break compared to CM (Supplementary Fig. 4). Different from CM which is easy to wrinkle and difficult to handle[44], BC-*g*-PNCl/CS-HAP can remain in a spreading state due to its appropriate rigidity and is also flexible enough to retain its original shape even after repeated bending and twisting (Fig. 3j).

## In vitro cytocompatibility, barrier function, and pro-osteogenic ability

Cytocompatibility, barrier function, and pro-osteogenic ability are essential for GBR membranes[45,46]. The results of CCK-8 assay (Fig. 4a), cytoskeleton arrangement (Supplementary Figs. 5 and 6), bone marrow mesenchymal stem cells (BMSCs) ingrowth (Supplementary Fig. 7) and L929s distribution (Supplementary Fig. 8) indicate good cytocompatibility of BC-*g*-PNCl/CS-HAP. The barrier function test (Fig. 4b, c and Supplementary Fig. 9) shows that penetrating cells of CM are obvious after 3 days and significantly increase after 7 days, while those in CS-HAP remain negligible. After simulated clinical immersion of membranes for 4 weeks, more fibroblasts penetrate CM, whereas almost no fibroblasts succeed in traversing through BC-*g*-PNCl/CS-

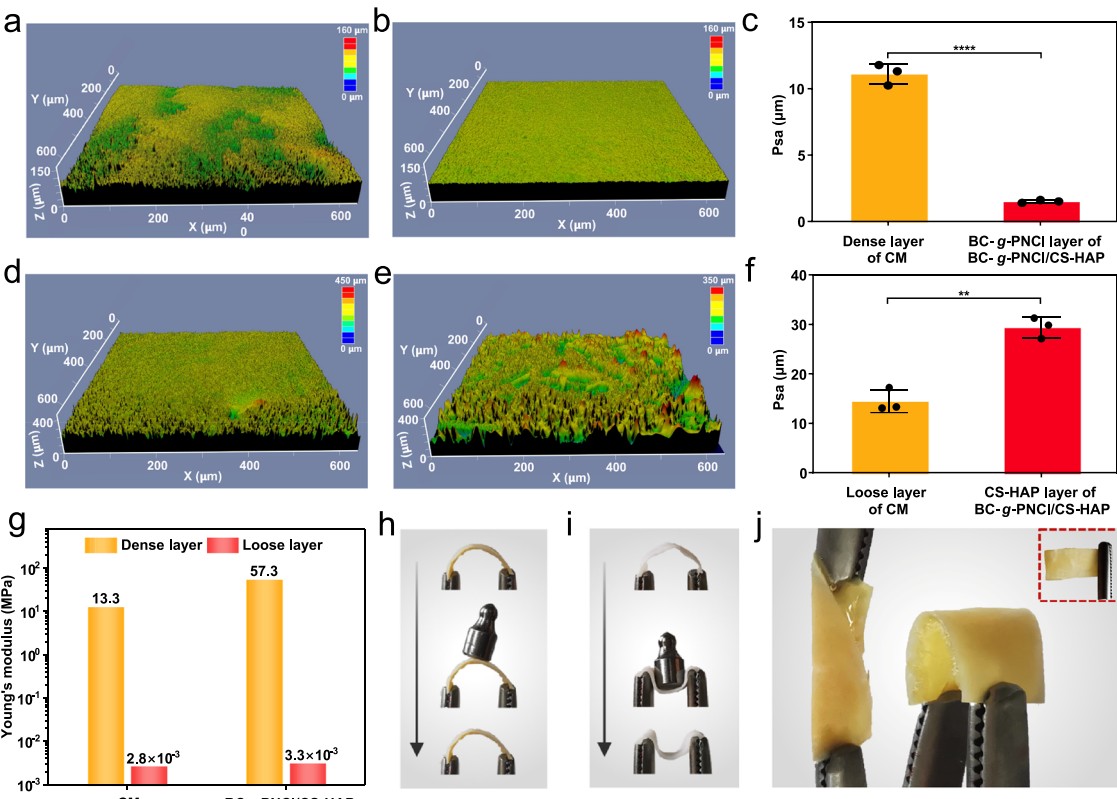

**Fig. 3 | Surface roughness and mechanical properties. a–c** Roughness of the dense layer of CM (**a**) and BC-*g*-PNCl layer of BC-*g*-PNCl/CS-HAP (**b**), and the corresponding roughness analysis (**c**) (*n* = 3 independent samples; Student's *t* test; **** two-tailed *P* < 0.0001; error bars = SD; data are presented as mean values ± SD). **d–f** Roughness of the loose layer of CM (**d**) and CS-HAP layer of BC-*g*-PNCl/CS-HAP (**e**), and the corresponding roughness analysis (**f**) (*n* = 3 independent samples;

Student's *t* test; ** two-tailed *P* = 0.0012; error bars = SD; data are presented as mean values ± SD). **g** AFM Young's moduli for CM and BC-*g*-PNCl/CS-HAP in the wet state. **h, i** Digital photos of BC-*g*-PNCl/CS-HAP (**h**) and commercial CM (**i**) before, during, and after the application of a load of 1 g in the wet state. **j** Digital photos of bending and twisting for BC-*g*-PNCl/CS-HAP (inset: digital photo after bending and twisting).

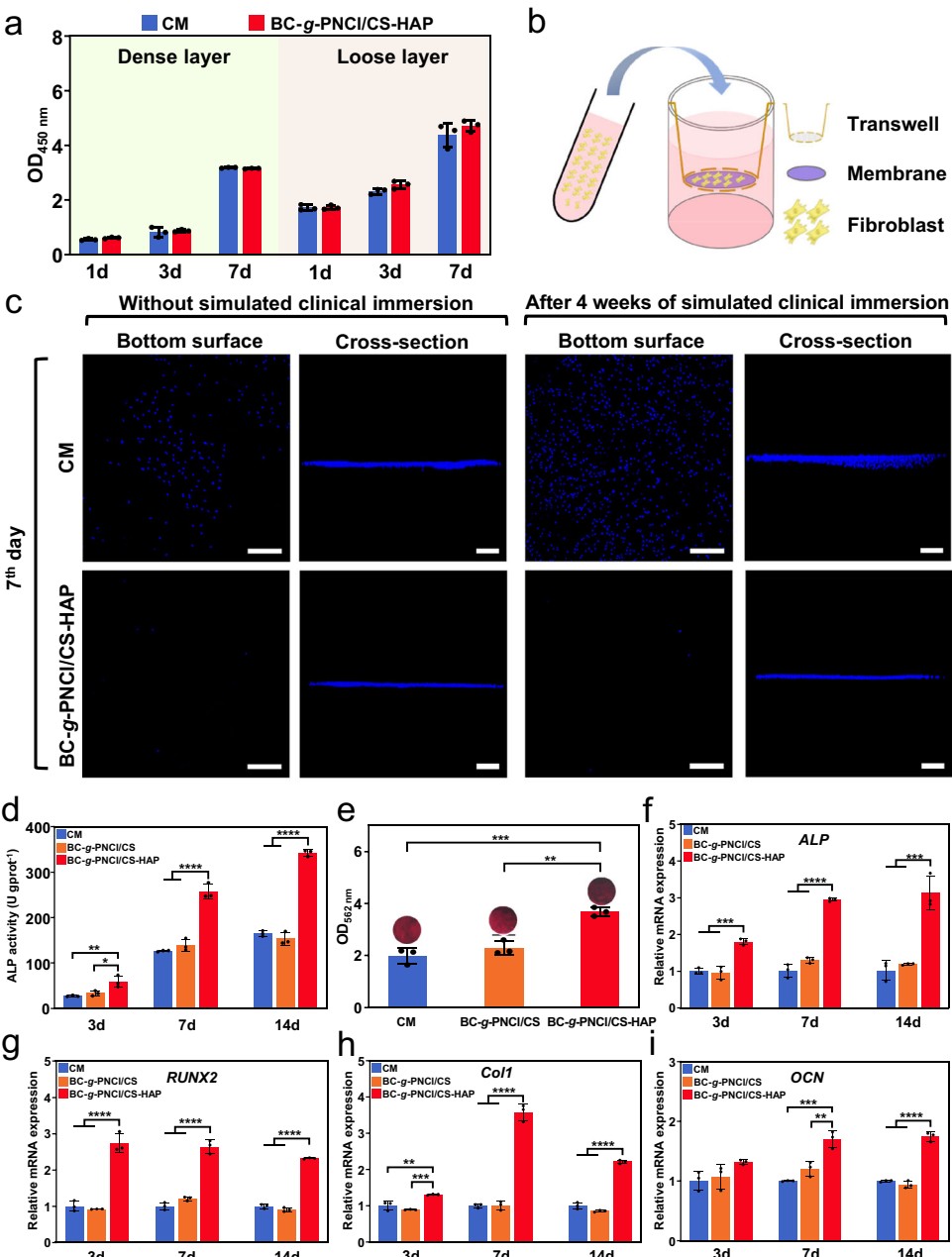

**Fig. 4 | In vitro cytocompatibility, barrier function, and pro-osteogenic ability.**
**a** Absorbances of the CCK-8 assay exhibiting the proliferation of L929s cultured on the dense layer and BMSCs cultured on the loose layer have no significant difference between CM and BC-*g*-PNCl/CS-HAP after 1, 3, and 7 days (*n* = 3 independent samples; Student's *t* test; two-tailed *P* = 0.088, 0.613, 0.056 in dense layer group and 0.938, 0.081, 0.294 in loose layer group for 1, 3 and 7 days compared with CM, respectively; error bars = SD; data are presented as mean values ± SD). **b** Schematic diagram showing the barrier function evaluation via transwell assay. **c** Fluorescent images of penetrated cells on the bottom surfaces and penetration depths after 7 days for CM and BC-*g*-PNCl/CS-HAP, exhibiting their barrier functions against

L929s before and after simulated clinical immersion for 4 weeks (blue for cellular nucleus, scale bars = 200 μm). **d** ALP activities of BMSCs cultured on the loose layers of samples for 3, 7, and 14 days. **e** Alizarin Red S staining of BMSCs cultured on the loose layers of samples for 21 days (red for calcium nodules). **f**–**i** RT-qPCR results of expression levels of osteogenic-related genes (*ALP*, *RUNX2*, *Col1*, and *OCN*) of BMSCs cultured on the loose layers of samples for 3, 7, and 14 days after osteogenic induction. **d**–**i** *n* = 3 independent samples; ANOVA followed by Tukey's multiple comparisons; *adjusted *P* < 0.05, **adjusted *P* < 0.01, ***adjusted *P* < 0.001, ****adjusted *P* < 0.0001; error bars = SD; data are presented as mean values ± SD.

HAP, which is expected to better protect the bone regeneration process. Moreover, the osteogenic assessments display that alkaline phosphatase (ALP) activity and content of calcium nodules in BMSCs cultured on the CS-HAP layer of BC-*g*-PNCl/CS-HAP always exhibit the highest (Fig. 4d, e). The as-mentioned BMSCs also exhibit the highest gene expression of *ALP*, *Runt-related transcription factor 2* (*RUNX2*), and *type I collagen* (*Col1*) on the 3rd, 7th, and 14th days, as well as *osteocalcin* (*OCN*) on the 7th and 14th days (Fig. 4f–i). Overall, our BC-*g*-

PNCl/CS-HAP can effectively support the ingrowth of BMSCs, barrier fibroblasts and activate osteogenesis.

**In vitro and in vivo multiple defenses against bacterial invasion**
Bacterial invasion after GBR surgery usually starts from the outside to the inside of GBR membranes, which should be blocked by the membrane. In SEM (Fig. 5a) and fluorescent images (Supplementary Fig. 10), both *S. aureus* and *Porphyromonas gingivalis* (*P. gingivalis*) pile on the

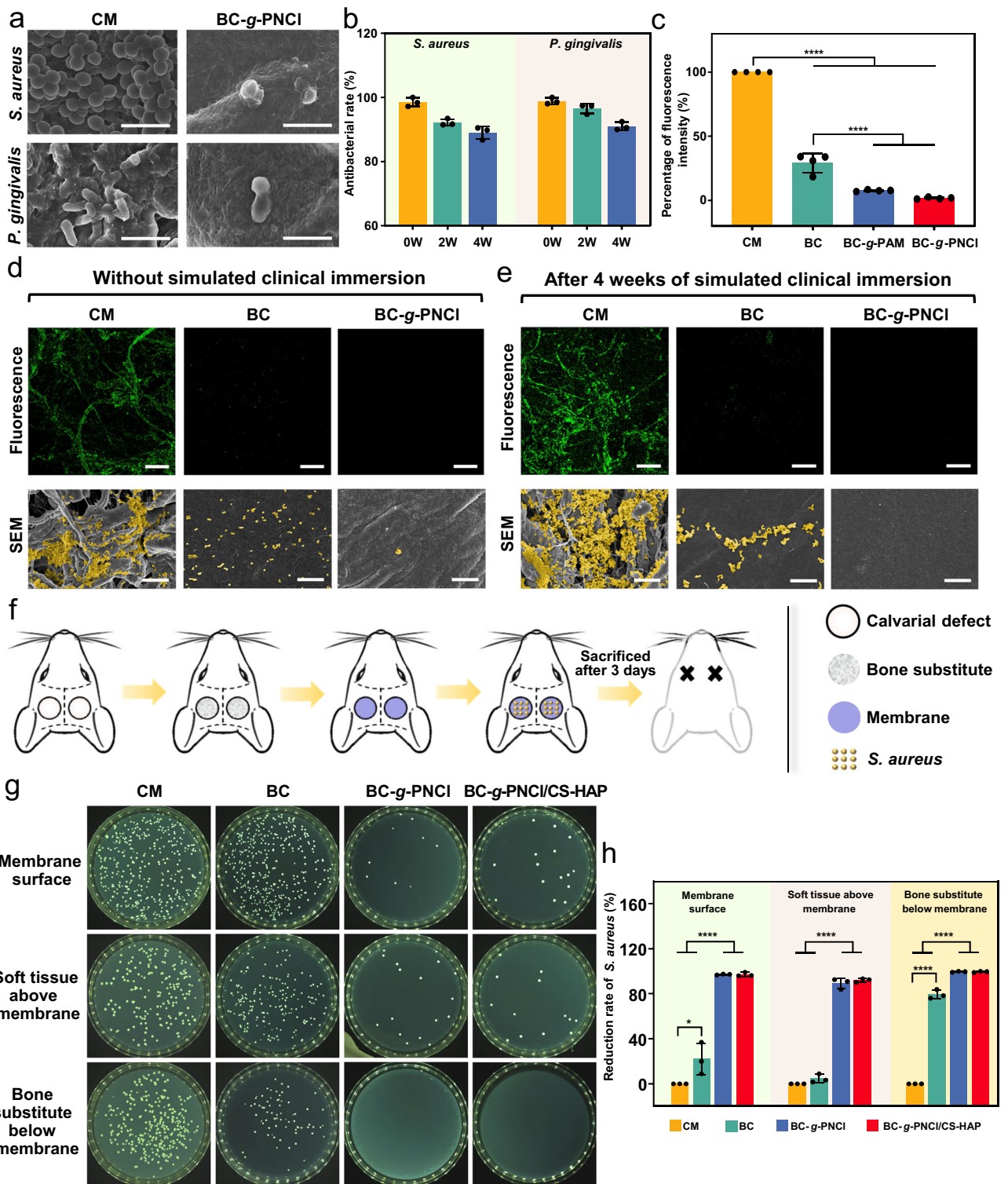

surface of CM, forming thick biofilms with intact cellular morphologies. In sharp contrast, only very few distorted and incomplete bacteria adhere to the surface of BC-*g*-PNCl. These results demonstrate BC-*g*-PNCl has anti-biofilm effects, which could be attributed to antibacterial and anti-adhesion performances.

As one of the most attractive bactericidal agents, N-halamines have been utilized in fields of biomedicine except for GBR membranes[47,48]. The release-killing rates of BC-*g*-PNCl against *S. aureus* and *P. gingivitis* in the medium are 73% and 69%, respectively (Supplementary Fig. 11), suggesting that the first defense can be

established by releasing antibacterial components before bacteria reach the membrane. When bacteria further adhere to the surface of BC-*g*-PNCl, 98% of *S. aureus* and 99% of *P. gingivalis* will be eliminated (Supplementary Fig. 12), providing the second defense. The first 4 weeks after GBR surgery are critical for soft tissue sealing and initial bone regeneration[6], so the duration of antibacterial defenses should ideally cover this period. As the storage time in artificial saliva increases, the contact-killing rates of BC-*g*-PNCl against *S. aureus* and *P. gingivalis* still remain at 89% and 91% after 4 weeks, respectively (Fig. 5b), indicating that BC-*g*-PNCl can maintain sufficient content of

**Fig. 5 | In vitro and in vivo multiple protections against bacterial invasion.**
**a** SEM images exhibiting the anti-biofilm effect against *S. aureus* and *P. gingivalis* on surfaces of CM dense layer and BC-*g*-PNCl (scale bars = 2 μm). **b** Quantitative measurements of antibacterial rates against *S. aureus* and *P. gingivalis* of BC-*g*-PNCl via CFU counting after samples are stored in artificial saliva for different duration times (*n* = 3 independent samples; error bars = SD; data are presented as mean values ± SD). **c** Anti-adhesion effect of CM dense layer, BC, BC-*g*-PAM, and BC-*g*-PNCl (*n* = 4 independent samples; ANOVA followed by Tukey's multiple comparisons; ****adjusted *P* < 0.0001, adjusted *P* = 0.173 between BC-*g*-PAM and BC-*g*-PNCl; error bars = SD; data are presented as mean values ± SD). **d**, **e** Fluorescent images (scale bars = 200 μm) and SEM images (scale bars = 10 μm) exhibiting the anti-penetration effect against *S. aureus* of CM and BC-*g*-PNCl before (**d**) and after (**e**) simulated clinical immersion for 4 weeks, indicating penetrated bacteria on the

bottom surface (green for live penetrated bacteria in fluorescent images, pseudo-yellow for penetrated bacteria in SEM images). **f** Schematic diagram showing the process of a short-term model of infected calvarial defects in rats to detect in vivo multiple protection against bacterial invasion. **g** Images of bacterial colonies formed by *S. aureus* survival on the surfaces of membranes (CM, BC, BC-*g*-PNCl, BC-*g*-PNCl/CS-HAP), soft tissues above the membranes, and bone substitutes below the membranes. **h** Reduction rates of *S. aureus* survival on the membrane surfaces, soft tissues above the membranes, and bone substitutes below the membranes, indicating the multiple protection effects of BC-*g*-PNCl and BC-*g*-PNCl/CS-HAP (*n* = 3 independent samples; ANOVA followed by Tukey's multiple comparisons; *adjusted *P* = 0.0222, ****adjusted *P* < 0.0001; error bars = SD; data are presented as mean values ± SD). SEM observation was repeated three times independently, yielding similar results.

antibacterial chlorine during the key stage of GBR to achieve long-lasting antibacterial defenses.

For general antibacterial biomaterials, the killed bacteria accumulating on the surface can reduce the bactericidal effectiveness[49,50] and promote the subsequent formation of biofilms[51,52]. Therefore, the anti-adhesion performance is particularly important for biomaterials. As shown in Fig. 5c and Supplementary Fig. 13, the fluorescence intensity reflecting adhered bacterial numbers for BC is only 29% of that for CM, resulting from the low surface roughness and nanoscale surface morphology of BC. The fluorescence intensity for BC-*g*-PAM further decreases by 72% compared with that for BC and is only 8% of that for CM, because of the reduced surface roughness (Supplementary Fig. 14) and surface hydration layer formed by amide groups of PAM and water[53,54]. As a result, just less than 2% of bacteria can adhere to BC-*g*-PNCl as compared with CM and most of the adhered bacteria are dead (Fig. 5c and Supplementary Fig. 13), as BC-*g*-PNCl continuously releases active chlorine and is eventually transformed into BC-*g*-PAM. Such good anti-biofilm capacities of BC-*g*-PNCl are also well supported by SEM images (Supplementary Fig. 15) and three-dimensional fluorescent images (Supplementary Fig. 16). The above results clearly indicate that the combined anti-adhesion function of BC backbones and dechlorination-generated amide groups can serve as the third reliable defense.

Inspired by the bacterial filter, we utilize the densely porous skeleton of BC-*g*-PNCl to block the remaining viable bacteria after the above antibacteria and anti-adhesion defenses that may penetrate the membrane through pores and infect the underlying bone substitutes, thus establishing the fourth defense. The bacterial penetration test (Fig. 5d) shows that after 3 days, a large number of bacteria arrive at the bottom surface of CM, while only scattered bacteria can reach that of BC and almost no bacteria successfully get to that of BC-*g*-PNCl. Even after 4 weeks of simulated clinical immersion, BC and BC-*g*-PNCl still maintain the above effects, whereas significantly more bacteria penetrate CM (Fig. 5e). Predictably, such a contrast will become sharper over time due to the high bacterial affinity of CM and the accelerated enzymatic degradation of collagen by bacteria[55,56] and triggered immune cells[57]. Therefore, BC-*g*-PNCl can act as a filter against bacteria and perfectly impede bacterial invasion because of its antibacterial, anti-adhesion, and anti-penetration properties. In case a small number of bacteria avoid the dense layer and approach the bone defect cavity from other directions, such as membrane edges and infected areas inside the alveolar bone, CS component in CS-HAP loose layer with a mild antibacterial effect from cationic amino groups[58] can display an antibacterial rate of 69% (Supplementary Fig. 17), thus forming the fifth defense.

The favorable in vitro results prompt us to further investigate the efficacy of BC-*g*-PNCl/CS-HAP on protection in more complex in vivo environments. We create a short-term model of the infected calvarial defect in rats to elucidate immediate protection of BC-*g*-PNCl/CS-HAP against bacterial invasion from the outside into the bone defect site (Fig. 5f). As displayed in Fig. 5g, h, bacteria on the surfaces of both BC-

*g*-PNCl and BC-*g*-PNCl/CS-HAP are dramatically reduced by 97% compared with that of CM. As for the soft tissues above membranes, compared to CM group, bacteria in BC-*g*-PNCl and BC-*g*-PNCl/CS-HAP groups are reduced by 89% and 92%, respectively (Fig. 5g, h). The above results demonstrate that the grafted N-halamine polymer chains can maintain high antibacterial rates in vivo. Moreover, for the bone substitutes below membranes that require strict protection, the number of bacteria in BC group is 79% lower than that in CM group (Fig. 5g, h). More importantly, bone substitutes in both BC-*g*-PNCl and BC-*g*-PNCl/CS-HAP groups are completely free of bacteria, confirming the ideal resistance to bacterial penetration and a sterile regeneration environment for the underlying bone substitutes. In a word, our BC-*g*-PNCl/CS-HAP can provide multiple protections for the surrounding soft tissues and underlying bone substitutes in a complex in vivo environment, thus providing an advantageous foundation for bone regeneration in the defect sites.

## In vivo pro-osteogenic ability

Two long-term models of rat calvarial defect are employed to further clarify the effects of BC-*g*-PNCl/CS-HAP on bone regeneration under normal and infected conditions (Fig. 6a). After 8 weeks of implantation, microcomputed tomography (micro-CT) data clearly show increased amounts and higher parameters of new bone formed in BC-*g*-PNCl/CS-HAP group than in CM group, no matter for the normal model or bacterial model (Fig. 6b−e). What is worth mentioning is that the regenerated bone of CM group in the bacterial model is much less than that in the normal model, indicating that the defect site in the bacterial model is most likely infected and bone regeneration is inhibited. In sharp contrast, the amount of newly formed bone in BC-*g*-PNCl/CS-HAP group in the bacterial model keeps the same level as that in the normal model, suggesting that BC-*g*-PNCl/CS-HAP can effectively block bacterial invasion and maintain a good microenvironment for bone regeneration. Moreover, hematoxylin and eosin (H&E) staining and Masson's trichrome staining (Fig. 6f) display that in the normal model, almost all defect spaces in BC-*g*-PNCl/CS-HAP group are filled with confluent mature bone with larger new bone area compared to CM group. In the bacterial model, calvarial defects in CM group are occupied with sporadic new bone, obvious fibrous connective tissues, and apparent inflammatory cells due to bacterial infiltration. In stark contrast, BC-*g*-PNCl/CS-HAP group still has plenty of dense new bone without inflammation, which displays the same tendency as the micro-CT results. The superior capability of BC-*g*-PNCl/CS-HAP to repair bone defects can be mainly attributed to the long-lasting multiple protection mechanisms of BC-*g*-PNCl dense layer to successfully block the ingrowth of surrounding fast-growing connective tissues and bacteria, together with the pronounced osteogenic ability of CS-HAP loose layer to promote bone formation. Furthermore, due to a slow and partially degradable characteristic, BC-*g*-PNCl/CS-HAP exhibiting reasonable resistance to external forces in a wet state has only a low mass loss of 31% after immersion in phosphate-buffered saline (PBS) for up to 12 weeks (Supplementary Fig. 18), ensuring long-lasting maintenance

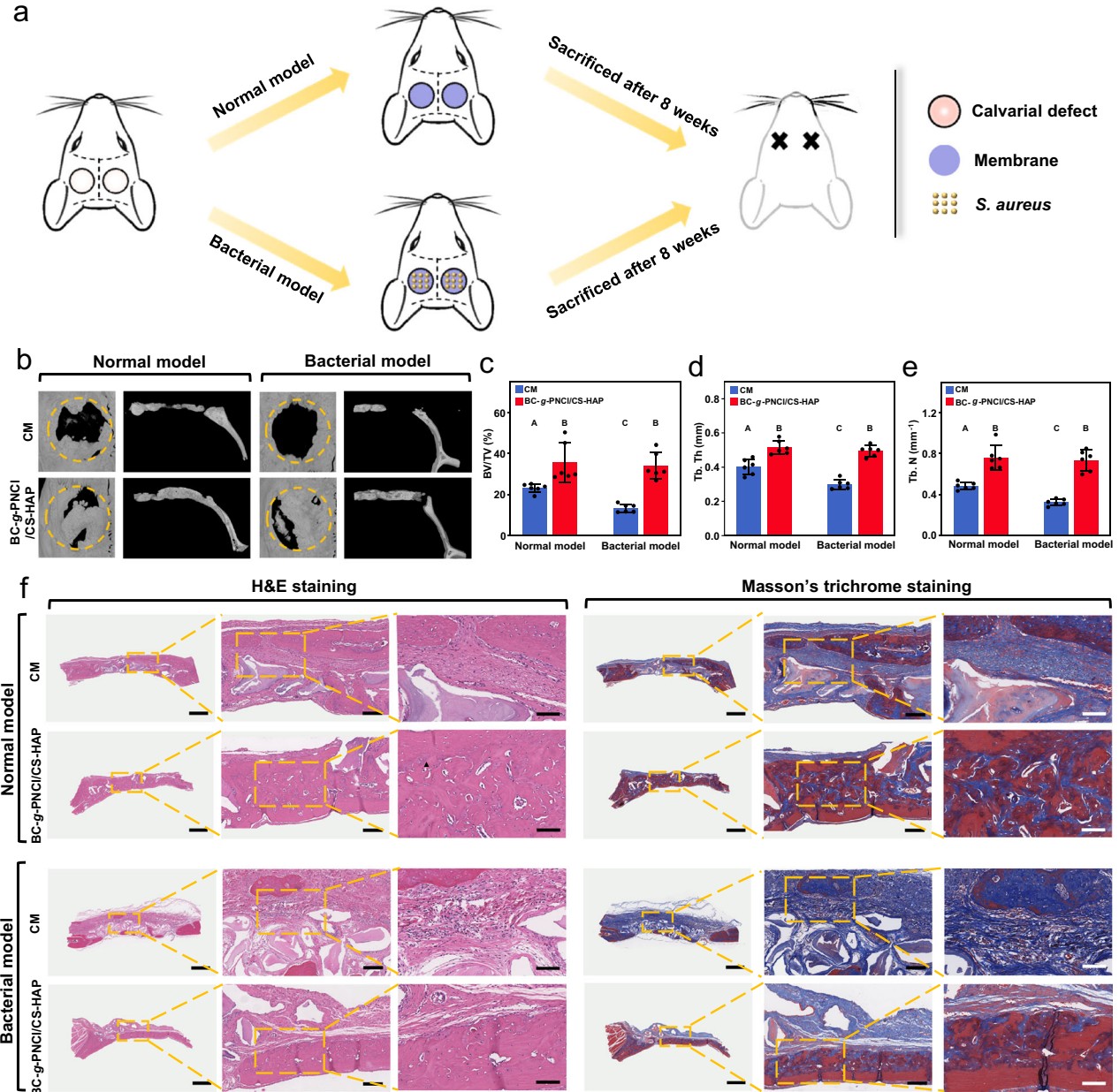

**Fig. 6 | In vivo pro-osteogenic ability in normal and bacterial models.**
**a** Schematic diagram showing the process of long-term models of rat calvarial defects to assess in vivo osteogenic ability under normal and bacterial conditions. **b** Micro-CT 3D reconstructions and coronal views of new bone inside bone defect sites after 8 weeks of implantation in both normal and bacterial models. **c**–**e** Quantitative analyses of bone parameters including BV/TV (**c**), Tb. Th (**d**), and Tb. N (**e**) of new bone after 8 weeks of implantation in both normal and bacterial models ($n = 6$ independent samples; ANOVA followed by Tukey's multiple comparisons; different letters mean adjusted $P < 0.05$ for CM or BC-*g*-PNCl/CS-HAP group among different models and within each model between groups; error bars = SD; data are presented as mean values ± SD). **f** Sections of specimens stained with H&E (left, middle, and right scale bars = 1 mm, 200 µm, and 100 µm, respectively) and Masson's trichrome (left, middle, and right scale bars = 1 mm, 200 µm, and 100 µm, respectively) of calvarial decalcified sections after 8 weeks of implantation in both normal and bacterial models.

of osteogenic space. This could also be beneficial for BC-*g*-PNCl/CS-HAP to achieve better osteogenic effects.

## Discussion

The oral cavity is one of the most complex regions in the body, possessing multiple contradictory factors such as the combination of soft and hard tissues, adjacency to internal and external environments, and richness in bacteria and cells[59–62]. However, the existing GBR membranes with a single structure or function cannot cope with the paradoxical conditions. Especially, they could fail to achieve satisfactory integration in terms of mechanical property, pore structure and

biological performance[5,7,11,20]. In this study, we demonstrate a spatio-temporally hierarchical GBR membrane by comprehensively integrating contradictory structures and properties. The spatiotemporally hierarchical design is manifested as the asymmetric design in terms of mechanics, pore structure, and biological characteristics. Firstly, BC-*g*-PNCl/CS-HAP harmoniously combines rigidity and flexibility that can withstand external pressure to maintain osteogenic spaces, bear bending and twisting, and conform to both soft and hard tissues. Secondly, in order to enable osteoblasts to grow without interference from soft tissue cells, BC-*g*-PNCl/CS-HAP is successfully designed to resist fibroblasts through small pores of the dense layer but promote

the ingrowth of osteogenesis-associated cells via large pores of the loose layer. Thirdly, HAP in the loose layer can activate BMSCs during both the early and late stages of osteogenesis and release $Ca^{2+}$ and $PO_4^{3-}$ to facilitate the formation of bone apatite[63]. More importantly, BC-*g*-PNCl/CS-HAP possesses multi-directional resistance to bacterial invasion through quintuple protections, including the long-lasting release-killing and contact-killing abilities of N-halamine, the combined anti-adhesion function of BC backbones and hairy polymer chains with in situ dechlorination-generated amide groups, the stable anti-penetration capacity of the nanoscale porous skeleton of BC-*g*-PNCl, and the mild antibacterial effect of CS, which could eventually create a sterile osteogenic microenvironment. In summary, our BC-*g*-PNCl/CS-HAP overcomes the shortcomings of the traditional GBR membranes, which enables us to propose a concept of an all-in-one GBR membrane. We believe our work provides an innovative direction for the development of GBR membranes as well as a typical paradigm for the design of complex biomaterials.

## Methods

### Fabrication of asymmetric porous bilayer membrane

BC membranes (BC-f0.3, Qihong Technology Co. Ltd, Guilin, China) were washed with distilled water and soaked in 75% ethanol/water solution with $N_2$ bubbling for 30 min. In all, 3% v/v KH570 (Yuanye Biotechnology Co. Ltd, Shanghai, China) and 1.5% v/v acetic acid were then added, followed by stirring the mixture at room temperature for 30 min and under 70 °C for 3 h to introduce C=C bonds onto the surface of BC. After cleaning with ethanol and distilled water, the resulting membranes were soaked in 80 mL distilled water with $N_2$ bubbling for 30 min, followed by adding 3.5 g acrylamide (Macklin, Shanghai, China) and 35 mg potassium persulphate (Sinopharm Chemical Reagent Co. Ltd, Shanghai, China) and heating for 5 h under 60 °C. BC-*g*-PAM products were obtained after washing with distilled water. To convert N-H bonds to N-Cl bonds, BC-*g*-PAM samples were soaked in a large excess of 1% NaClO (Macklin, Shanghai, China) aqueous solution for 12 h at 4 °C and washed with distilled water. The excess water on the membrane surface was lightly wiped off, yielding the wet BC-*g*-PNCl dense layer.

In total, 0.5 g CS (Aladdin, Shanghai, China) and 0.25 g HAP (Macklin, Shanghai, China) were added into 25 mL acetic acid aqueous solution (2% v/v). Subsequently, the CS was slightly crosslinked for 5 min at room temperature after adding 30 μL glutaric dialdehyde aqueous solution (50% v/v; Aladdin, Shanghai, China). The as-obtained solution was then evenly dropped onto the surface of the above wet BC-*g*-PNCl dense layer, and the resulting membrane was frozen with liquid nitrogen and rapidly soaked in 0.3 M cold NaOH ethanol solution (−20 °C) several times to remove acetic acid. The as-obtained frozen crude product was quickly washed with distilled water several times and then freeze-dried, yielding the asymmetric porous bilayer membrane BC-*g*-PNCl/CS-HAP. According to the above procedures, BC-*g*-PNCl/CS sample was prepared without adding HAP, and CS loose sample was also fabricated by dropping CS solution into a culture dish rather than onto the surface of the above wet BC-*g*-PNCl dense layer. Additionally, the as-received BC membrane was rinsed several times with distilled water and freeze-dried, producing BC sample; the above wet BC-*g*-PAM and BC-*g*-PNCl products were freeze-dried, yielding BC-*g*-PAM and BC-*g*-PNCl samples, respectively. Considering Bio-Gide® collagen membrane (Geistlich, Switzerland) has been widely recognized as the gold clinical standard in GBR therapy, it was also used as a control sample.

### Structural and mechanical characterization

FTIR spectra were recorded by FTIR spectrometer (Nicolet 6700, Thermo Scientific, USA). XPS spectra were recorded by X-ray photoelectron spectrometer (Nexsa, Thermo Fisher, USA). Elemental mappings were conducted utilizing SEM (ΣIGMA 300, Zeiss, Germany) combined with an energy dispersive spectrometer (XFlash 6|30, Bruker, Germany). The bilayer morphology of the membrane side was visualized by an ultra-depth three-dimensional microscope (VHX-1000C, Keyence, Japan). BC-*g*-PNCl/CS-HAP was soaked in water and shaken for 20 s by a vortex oscillator (MX-S, Dragon, Beijing, China) to verify the tight connectivity between the dense layer and the loose layer. The sizes of pores in the membrane surfaces were analyzed by SEM (S-4800, Hitachi, Japan). During in vitro simulated clinical immersion, membranes were immersed in PBS at 37 °C, and the surface morphologies of the dense layers of membranes were detected using SEM after immersion for 4 weeks. The roughness was assessed through the confocal laser scanning microscope (CLSM; LSM 700, Zeiss, Germany). Young's moduli of BC-*g*-PNCl/CS-HAP and collagen membrane in a wet state were measured by atomic force microscopy (AFM; Dimension Fastscan, Bruker, Germany) in the peak force quantitative nanomechanics mode and then analyzed utilizing the Derjaguin−Muller−Toporov model. Before the investigation of mechanical properties (Young's modulus, tensile, loading, bending, and twisting), the samples were sterilized with ethylene oxide and soaked in saline for at least 1 h, and the excess saline on the membrane surface was lightly wiped off.

### Weight loss testing

BC-*g*-PNCl/CS-HAP membranes were randomly allocated into eight groups ($n = 3$) after drying based on the total measurement time. Each membrane was weighed and recorded as $m_0$ ($40 \pm 10$ mg) and was then immersed into 5 mL of PBS (pH = 7.4) at 37 °C within a shaking apparatus. The PBS solution was replaced every 3 days. At each predetermined time point, each membrane in one group was taken out from PBS, rinsed 3 times with distilled water, dried for 24 h at 60 °C, weighted and recorded as $m_1$. The weight loss was calculated according to Eq. (1):

$$\text{Weight loss percentage (\%)} = \frac{m_0 - m_1}{m_0} \times 100\% \tag{1}$$

### Cytocompatibility evaluation

The cytocompatibility of BC-*g*-PNCl/CS-HAP was evaluated via CCK-8 (Dojindo, Tokyo, Japan) and fluorescent staining with CM as control. L929 fibroblasts (Cat. No. GNM28, National Collection of Authenticated Cell Cultures, Shanghai, China) were seeded onto the dense surfaces of samples, while BMSCs isolated from 2-week-old Sprague-Dawley male rats were cultivated on the loose surfaces. On the 1st, 3rd, and 7th days of cultivation, 200 μL of CCK-8 reagent was added to each well and incubated for 1 h. The optical density (OD) of the supernatant was measured through a microplate reader (Epoch 2, BioTek, USA) under 450 nm. The morphologies of BMSCs cultured on the loose layer and L929 fibroblasts cultured on the dense layer for 7 days were detected. The distributions of BMSCs cultured on the loose layer and L929 fibroblasts cultured on the dense layer for 3 and 7 days were also observed. The samples were fixed by 4% paraformaldehyde (Biosharp, Hefei, China) and permeabilized in 0.5% Triton X-100. Cells on the samples were stained with Actin-Tracker Green (Beyotime, Shanghai, China) and 4',6-diamidino-2-phenylindole (DAPI; Beyotime, Shanghai, China), and then visualized by CLSM (FV3000, Olympus, Japan).

### Barrier function assessment

The primary function of GBR membranes is to stably prevent the infiltration of soft tissue cells from the outside to the inside. To detect the barrier function of the dense layer against fibroblasts, both CM and BC-*g*-PNCl/CS-HAP were stored in sterilized PBS for different periods (0 and 4 weeks). At each time point, 20 μL L929 fibroblasts suspension was seeded onto the top dense layer surface of each sample sealed into a transwell, and 1 mL media was then added after 2 h to ensure an equal initial cellular density. After culturing for 3 and 7 days, cells on both

layers were stained with DAPI and observed with CLSM (FV3000, Olympus, Japan).

## In vitro osteogenic-related testing

CM, BC-*g*-PNCl/CS, and BC-*g*-PNCl/CS-HAP were used in experiments. The osteogenic abilities of BMSCs on loose surfaces were assessed by ALP activity, calcium deposition, and mRNA expression levels of osteogenic markers. The ALP activity of cells was detected on the 3rd, 7th, and 14th days of osteogenic induction with an ALP Assay Kit (Jiancheng, Nanjing, China). At every point in time, samples were washed with PBS and lysed with 1% Triton X-100 overnight at 4 °C. Cell lysates were then transferred to a 96-well plate and ALP activities were quantitatively measured. Calcium phosphate formation was examined utilizing Alizarin Red S staining. On the 21st day of osteogenic induction, BMSCs on each sample were fixed with 4% paraformaldehyde, stained by Alizarin Red solution (Cyagen, Suzhou, China), and then viewed by stereomicroscope (MZ10F, Leica, Germany). Calcium content was analyzed semi-quantitatively with a microplate reader (Epoch 2, BioTek, USA) at 562 nm. In terms of osteogenesis-associated genes including *ALP*, *RUNX2*, *Col1*, and *OCN*, their expression levels were estimated at the 3rd, 7th, and 14th days after osteogenic induction by reverse transcription-quantitative polymerase chain reaction (RT-qPCR). Total RNA was isolated from BMSCs using an RNA-Quick Purification Kit (Yishan, Shanghai, China). 500 ng RNA was reverse transcribed to cDNA by PrimeScript™ RT Reagent Kit (Takara, Dalian, China). RT-qPCR was performed on a Real-Time PCR System (LightCycler 96, Roche, Switzerland). The primer sequences of tested genes were displayed in Supplementary Table 1 and *glyceraldehyde 3-phosphate dehydrogenase* (*GAPDH*) was employed as the housekeeping gene. Relative expressions of the above genes were calculated using the $2^{-\Delta\Delta Ct}$ method, normalizing to the levels of reference gene *GAPDH*.

## Anti-biofilm effect assessment

The anti-biofilm effects of BC-*g*-PNCl were evaluated with aerobic *S. aureus* (ATCC25923, Guangdong Microbial Culture Collection Centre, Guangzhou, China) and anaerobic *P. gingivalis* (ATCC33277, Guangdong Microbial Culture Collection Centre, Guangzhou, China) through SEM and fluorescence investigations. *S. aureus* was placed in Luria Bertani (LB) growth medium at 37 °C and *P. gingivalis* was placed in brain heart infusion (BHI) broth in an anaerobic environment (80% $N_2$, 10% $H_2$ and 10% $CO_2$) at 37 °C. CM and BC-*g*-PNCl were used in the experiments. 300 μL of suspension containing $10^6$ colony-forming units (CFU) mL$^{-1}$ *S. aureus* or *P. gingivalis* was seeded onto each sample and incubated for 24 h. For SEM imaging, the samples were fixed with 2% glutaraldehyde overnight at 4 °C, sequentially dehydrated with gradient ethanol, covered with gold, and investigated with SEM (S-4800, Hitachi, Japan). For fluorescent imaging, the samples were rinsed with PBS, stained with LIVE/DEAD *Bac*Light Bacterial Viability Kit (Molecular Probes Inc., Eugene, USA) for 15 min in the dark and observed via CLSM (FV3000, Olympus, Japan).

## Antibacterial effect assessment

The in vitro antibacterial ability of BC-*g*-PNCl was assessed using *S. aureus* and *P. gingivalis* through the plate counting method by determining the amount of CFU on agar plates. BC-*g*-PNCl and BC were used in the experiments. To accurately evaluate the antibacterial performances of N-halamine, we calculated the antibacterial rate of BC-*g*-PNCl with BC as the control group. 300 μL of bacterial suspension ($10^6$ CFU mL$^{-1}$) was seeded onto each sample and incubated for 16–18 h. To detect the release-killing ability of BC-*g*-PNCl, the corresponding culture medium was diluted with PBS 250,000-fold for *S. aureus* and 120,000-fold for *P. gingivalis*. 20 μL of the diluted bacterial suspension was seeded and spread over agar plates evenly. Specifically, *S. aureus* suspension was inoculated onto LB agar plates and *P. gingivalis* suspension was inoculated over BHI agar plates containing 10% sheep blood. To

determine the contact-killing ability of BC-*g*-PNCl, adhered bacteria were isolated into 1 mL PBS from each sample by ultrasonic vibration (300 W, 40 kHz, 3 min). The bacterial suspension was diluted with PBS, in which *S. aureus* was diluted 2500 times and *P. gingivalis* was diluted 1000 times, and then spread on the corresponding agar plate.

To test the long-lasting contact-killing ability of BC-*g*-PNCl, BC-*g*-PNCl samples, and their BC control samples were stored in sterilized artificial saliva for certain durations (0, 2, and 4 weeks). At each time point, samples were cultured with bacterial suspension, followed by the detection of contact-killing ability as described above. Release- and contact-type antibacterial rates were both calculated by Eq. (2):

$$\text{Antibacterial rate}\,(\%) = \frac{\text{CFU of BC-CFU of BC-}g\text{-PNCl}}{\text{CFU of BC}} \times 100\% \qquad (2)$$

To evaluate the antibacterial activity of CS against *S. aureus*, each CS loose sample was immersed in 300 μL bacterial suspension containing $10^6$ CFU mL$^{-1}$ *S. aureus* and incubated at 37 °C, 50 rpm for 12 h. The bacterial suspension without a sample was set as the blank group. The culture medium was then diluted 200,000-fold and seeded onto LB agar plates. The antibacterial rate of CS was calculated by Eq. (3):

$$\text{Antibacterial rate}\,(\%) = \frac{\text{CFU of blank} - \text{CFU of CS}}{\text{CFU of blank}} \times 100\% \qquad (3)$$

## Anti-adhesion effect assessment

The anti-adhesion effect against bacteria of various components of BC-*g*-PNCl dense layer was evaluated using *S. aureus* through fluorescent staining and SEM. CM, BC, BC-*g*-PAM, and BC-*g*-PNCl were used in the experiments. 300 μL of bacterial suspension ($10^6$ CFU mL$^{-1}$) was inoculated onto each sample and incubated for 24 h. Bacteria attached to samples were stained using LIVE/DEAD *Bac*Light Bacterial Viability Kit and observed through CLSM, and bacteria on samples were also detected via SEM as described previously. The fluorescence intensity reflected the number of attached bacteria. The integrated density (IntDen) representing fluorescence intensity in images was quantitively analyzed using the Image J software (v1.6.0, National Institute of Health, Bethesda, USA). The percentages of fluorescence intensity representing the anti-adhesion effect for experiment groups (BC, BC-*g*-PAM, and BC-*g*-PNCl) were calculated according to Eq. (4):

$$\text{Percentage of fluorescence intensity}\,(\%) = \frac{\text{IntDen of experiment group}}{\text{IntDen of CM}} \times 100\%$$

$$(4)$$

## Anti-penetration effect assessment

Bacterial intrusion should be stably obstructed by the dense layer of GBR membranes to prevent infection. To detect the ability to resist bacterial penetration, CM, BC, and BC-*g*-PNCl were stored in sterilized PBS. After 0 or 4 weeks of storage, 20 μL of *S. aureus* suspension ($10^7$ CFU mL$^{-1}$) was dropwise added onto the top dense surface of each sample sealed into a transwell and 1 mL media was added after 2 h to ensure that the initial amount of bacteria on different surfaces was as equal as possible. After 3 days of culture, samples were stained with LIVE/DEAD *Bac*Light Bacterial Viability Kit, and the penetrated bacteria in the bottom surfaces were observed by CLSM. Samples were fixed and sequentially dehydrated, and their bottom surfaces were conductively coated and investigated with SEM (ΣIGMA 300, Zeiss, Germany). *S. aureus* on the SEM images were mapped with pseudo-color.

## In vivo evaluation of protective and osteogenic properties

Male Sprague-Dawley rats (200–250 g, 6–7 weeks) were used for in vivo experiments under the permission of the Institutional Animal

Care and Use Committee of Sun Yat-sen University (No. SYSU-IACUC-2022-001183). To construct the critical-sized calvarial defect model, parietal bones were exposed and full-thickness round defects (5.5 mm in diameter) were created via a trephine bur on bilateral sides of the skull in each rat.

To analyze the immediate protection of our membranes against bacterial invasion, we created a short-term model of infected calvarial defects. Briefly, defect cavities were filled with an equal amount of bone substitutes (Bio-Oss, Geistlich, Switzerland) and covered by CM, BC, BC-*g*-PNCl, or BC-*g*-PNCl/CS-HAP. To mimic the clinical situation of external bacterial invasion after the exposure of GBR membranes, *S. aureus* suspension (10 μL, $10^8$ CFU mL$^{-1}$) was added onto the top dense surface of each membrane, and incisions were tightly sutured after 5 min. Rats were sacrificed after 3 days. Membranes, soft tissues above, and bone substitutes below were all collected in the biosafety cabinet. Bacteria adhered to each sample mentioned above were ultrasonically detached in 1 mL PBS solution. Bacteria suspensions of membrane surfaces, soft tissues above membranes, and bone substitutes below membranes were then diluted 200-fold, 50-fold, and 0-fold, respectively, and seeded on plates as previously described. The reduction rates of *S. aureus* in experiment groups (BC, BC-*g*-PNCl, and BC-*g*-PNCl/CS-HAP) were calculated by Eq. (5):

$$\text{Reduction rate of } S. aureus\,(\%) = \frac{\text{CFU of CM} - \text{CFU of experiemnt group}}{\text{CFU of CM}} \times 100\%$$

(5)

To study the effect of our membranes on bone regeneration under normal and infected conditions, we created another two long-term models including a normal model and a bacterial model. In brief, the generated defect cavities were covered with CM and BC-*g*-PNCl/CS-HAP, respectively. Incisions in the normal model were immediately sutured, while those in the bacterial model were sutured after *S. aureus* suspension (2 μL, $10^7$ CFU mL$^{-1}$) was added onto the dense surface of each membrane. After 8 weeks, all rats were sacrificed and calvariums containing the defect regions were removed and immersed into 4% paraformaldehyde for 48 h. Micro-CT (SkyScan 1276, Bruker, Germany) scanning was performed to detect the newly formed bones in defect cavities, and subsequent 3D reconstructions were obtained via CTvox software (v.3.3.0, Bruker, Karlsruhe, Germany). The analyses of bone volume fraction (BV/TV), trabecular thickness (Tb. Th), and trabecular number (Tb. N) were then performed to estimate formations of new bones. To further investigate the osteogenesis capability of BC-*g*-PNCl/CS-HAP, histological analyses of newly formed bones were conducted. The fixed specimens were decalcified by 10% EDTA, dehydrated in an ascending ethanol gradient, embedded in paraffin and sectioned coronally into 4-μm-thick slices. Sections were subsequently stained with H&E and Masson's trichrome.

### Statistical analysis
The data were expressed as the mean values ± standard deviation (SD). Statistical analysis was conducted by GraphPad Prism 8 software with Student's *t* test, one-way analysis of variance (ANOVA) followed by Tukey's multiple comparisons for data with homogeneity of variance, and Kruskal–Wallis's nonparametric test for data with heterogeneity of variance. $P < 0.05$ was regarded as statistically significant.

### Reporting summary
Further information on research design is available in the Nature Portfolio Reporting Summary linked to this article.

## Data availability
Data supporting the results of this study are available in the article and its supplementary information. All data underlying this study are accessible from the corresponding authors upon request.

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

## Acknowledgements

This work was supported by the projects of Guangdong Major Project of Basic and Applied Basic Research, National Natural Science Foundation of China (82071158, 82271031, and 51925308), Fundamental Research Funds for the Central Universities, Sun Yat-sen University (23yxqntd002), Guangdong Basic and Applied Basic Research Foundation (2022A1515012600 and 2021A1515010364), GDPH Supporting Fund for Talent Program (KY012021209) and National Key Research and Development Program of China (2022YFA1304000).

## Author contributions

D.W. and Y.L. supervised the project. S.W. and M.Z. conceived the underlying idea. S.W., S.L., and Z.C. designed the research and carried out the experiment. Q.L., L.L., W.L., Z.H., W.H., and G.L. performed the statistical analysis. S.W., S.L., and Z.C. wrote the manuscript. All the authors contributed to discussing and revising the manuscript.

## Competing interests

The authors declare no competing interests.
