## [Peer Review File · Nature Communications]

Reviewers' Comments:

Reviewer #1:

Remarks to the Author:

I appreciate in considering me to review this manuscript.

However, I am considering this manuscript to be accepted since the authors revised it, improving with my appointments (major review) below.

This paper is relevant to provide new directions for the development of antimicrobial regenerative membranes as well as a typical paradigm for the design of complex biomaterials, as has been concluded by the authors.

Although the barrier system presented in this article is a guideline for developing new therapeutic approaches, the authors must discuss the clinical relevance and predictability of a non-absorbable medical device since the regulations for medical devices have become increasingly demanding concerning the lifetime of implantable products, culminating in the trend is the use of absorbable implantable products.

Note: Bacterial cellulose is a biopolymer non-absorbable.

Another point the authors need to discuss in the article is, with advances in 3D-printed absorbable medical devices for filling and reconstruction of bone defects in GBR, what would be the clinical relevance of such a stiff membrane? The authors need to discuss further whether, for a GBR, only using this membrane would be enough. Wouldn't it be necessary to use bone grafts?

The authors must explain better regarding in vivo results. Owing to the performance, the BC-g-PNCl/CS-HAP membrane is better than the collagen membrane, is only not the composition of the membranes but also the differences between thickness (not reported), degradable mechanism of the products; in other words, there are more critical factors involved in these in vivo results that favored higher space maintenance and consequently a better outcome to bone neoformation. Please, discuss these results more.

The authors concluded, "...our BC-g-PNCl/CS-HAP breaks through the traditional concept that the GBR membrane can only shield soft tissue cells and proposes a new concept of all-in-one GBR membranes."

However, N-halamine for coatings and grafting is not a new idea; there are patents and other articles that use this concept. Furthermore, bacterial cellulose + chitosan in the way composite membrane with or without hydroxyapatite has several articles in the literature. Then, the authors must better defend the concept of the new idea and process.

Reviewer #2:

Remarks to the Author:

In this study, the authors fabricated novel spatiotemporally hierarchical guided bone regeneration (GBR) membranes by rational bilayer integration of densely porous N-halamine functionalized bacterial cellulose nanonetwork facing the gingiva and loosely porous chitosan-hydroxyapatite composite micronetwork facing the alveolar bone. The BC-g-PNCl/CS-HAP membrane has a mechanically matched space maintenance capacity toward gingiva, continuously blocks fibroblasts, and prevents bacterial invasion with multiple mechanisms including release-killing, contact-killing, anti-adhesion, and nanopore-blocking. Even though the findings are interesting, the result and discussion part is not well focused. Some data should be further analyzed and even more data are needed to improve this manuscript. The following issues need to be clarified.

1. The introduction is not interesting enough, and it should be rewritten to highlight the importance and novelty of this work.
2. Why choose chitosan to fabricate asymmetric porous bilayer BC-g-PNCl/CS-HAP membrane rather than biological materials? What is the function of it?
3. How about degradation rate of BC-g-PNCl/CS-HAP membrane? The degradation of BC-g-PNCl/CS-HAP should be provided.
4. As one of the most attractive bactericidal agents, N-halamines have been utilized in fields of

biomedicine except for GBR membranes. The release-killing rates of BC-g-PNCl against *S. aureus* and *P. gingivitis* in the medium are 73% and 69%, respectively (Fig. S9), suggesting that the first defense can be established by releasing anti-bacterial components before bacteria reach the membrane. What are the anti-bacterial components? Is it N-halamines? The study of anti-bacterial components release should be provided.

5. Why choose calvaria defect model for evaluation of bone regeneration in vivo, rather than bone defect model in oral cavity.

6. Immunohistochemical analysis of in vivo osteogenesis (Col-1 or BMP-2) would be beneficial.

7. Discussion is unfocused and too easy; authors should include the novelty of the present study compared to other studies in the discussion part as there are many studies on GBR membranes.

Reviewer #3:

Remarks to the Author:

In this manuscript, the authors have successfully developed a novel spatiotemporally hierarchical guided bone regeneration (GBR) membrane. Different from the existing GBR membranes, this new membrane integrates densely porous N-halamine functionalized bacterial cellulose nanonetwork facing the gingiva and loosely porous chitosan-hydroxyapatite composite micro-network facing the alveolar bone. Therefore, the new GBR membrane asymmetrically combines stiffness and flexibility, ingrowth barrier and ingrowth guiding, as well as anti-bacterial and cell-activation, which have been verified by comprehensive and reasonable in vitro and in vivo experiments. The interesting results move forward the development of GBR membrane and also provide a typical paradigm for the design of complex biomaterials. Overall, this research has a high novelty and clinical value, and the manuscript is well-written. So I suggest a minor revision after addressing the following concerns:

1. The authors should provide a more detailed description of the shortcomings of existing GBR membranes in clinical applications to highlight the clinical value of their study in the introduction section.

2. Since graft density is an important chemical structural parameter, the authors should provide the content of PAM in BC-g-PAM.

3. For the CS-HAP loose layer, the authors only characterized the presence of calcium and phosphorus in HAP by elemental mapping. They should further provide the presence of chitosan in the CS-HAP loose layer.

4. How were the Young's moduli in Fig. 3g obtained? The details of testing and analysis should be added.

5. For the mechanical properties, the authors only measured Young's moduli and performed bending and twisting tests. It is recommended to detect the tensile properties further.

6. For the biocompatibility tests, the authors only conducted in vitro studies. How is the biocompatibility of the GBR membrane in vivo? Why did they not carry out the in vivo biocompatibility experiments?

7. The authors pointed out that the loose layer was favorable for osteoblast proliferation. What about the effect of dense surfaces on fibroblast proliferation?

8. The authors carried out anti-biofilm experiments to compare the adhesion of bacteria on dense surfaces. It is recommended to more visually display the effect of dense layers on biofilm thickness, similar to the 3D cross-section display of cell penetration in barrier experiments.

9. How is the degradation property of the GBR membrane?

10. What is the bactericidal mechanism of the released antibacterial agent? Is it harmful to cells?

11. The authors mentioned that the prepared GBR membrane was spatiotemporally hierarchical. Relevant experiments or discussions are required to explain this property.

Response to Referees' Comments

Reviewer #1

I appreciate in considering me to review this manuscript. However, I am considering this manuscript to be accepted since the authors revised it, improving with my appointments (major review) below.

Thanks a lot for taking valuable time to review our manuscript. We really appreciate your constructive suggestions for enriching our work. We have carefully addressed all your concerns in our revised manuscript and provided the following point-to-point response.

1. This paper is relevant to provide new directions for the development of antimicrobial regenerative membranes as well as a typical paradigm for the design of complex biomaterials, as has been concluded by the authors.

Although the barrier system presented in this article is a guideline for developing new therapeutic approaches, the authors must discuss the clinical relevance and predictability of a non-absorbable medical device since the regulations for medical devices have become increasingly demanding concerning the lifetime of implantable products, culminating in the trend is the use of absorbable implantable products.

Note: Bacterial cellulose is a biopolymer non-absorbable.

Thanks for your encouraging comments and kind suggestions. Clinically, in order to achieve long-term and stable space maintenance, non-absorbable membranes are often chosen when dealing with severe or extensive bone defects. They can be passingly removed when cutting through the gingiva during the subsequent implant surgery. However, materials of non-absorbable membranes commonly used in clinical practice mainly include titanium and polytetrafluoroethylene. The former is too rigid and the latter exhibits poor performance in tissue integration, which is prone to cause soft tissue dehiscence and bacterial infection. Meanwhile, their lack of biological functions makes them difficult to achieve ideal bone augmentation effects (*Tissue Eng. Part B Re.* 2023,

0040, DOI: 10.1089/ten.teb.2023.0040). In sharp contrast, our BC-g-PNCl/CS-HAP can overcome the aforementioned shortcomings of existing non-absorbable membranes. Specifically, BC-g-PNCl/CS-HAP harmoniously combining rigidity and flexibility can withstand external pressure to maintain osteogenic spaces, bear bending and twisting, and conform to both soft and hard tissues. In order to harmonize the rapid proliferation of external fibroblasts with the slow migration of internal osteogenesis-associated cells, BC-g-PNCl/CS-HAP is successfully designed to resist fibroblasts through small pores of the dense layer but promote the ingrowth of BMSCs via large pores of the loose layer. Meanwhile, benefiting from the bone biomimetic composites in its loose layer, BMSCs can be activated during both the early and late stages of osteogenesis. Remarkably, BC-g-PNCl/CS-HAP possesses omni-directional resistance to bacterial invasion through quintuple protections, including the long-lasting release-killing and contact-killing ability of N-halamine, the combined anti-adhesion function of BC backbones and hairy polymer chains with in situ dechlorination-generated amide groups, the stable anti-penetration capacity of the nanoscale porous skeleton of BC-g-PNCl, and the mild anti-bacterial effect of CS. In summary, the performance of BC-g-PNCl/CS-HAP is perfectly matched to the needs of clinical applications.

2. Another point the authors need to discuss in the article is, with advances in 3D-printed absorbable medical devices for filling and reconstruction of bone defects in GBR, what would be the clinical relevance of such a stiff membrane? The authors need to discuss further whether, for a GBR, only using this membrane would be enough. Wouldn't it be necessary to use bone grafts?

Thanks for your nice comment. Within the field of bone tissue regeneration engineering, bone defect areas can be filled and reconstructed with 3D-printed absorbable materials. However, the proliferation rate of soft tissues (e.g., epithelium, connective tissue) is much higher than that of bone tissue. This inherent difference causes the overlying soft tissue to quickly grow into the filling material and osteogenic space, which hinders bone repair. To address this issue, the guided bone regeneration

(GBR) technique has been proposed and widely used for augmenting alveolar bone and treating peri-implant bone deficiencies. The key principle of GBR is utilization of a membrane covering above filling material to prevent ingrowth of rapidly proliferating epithelium and connective tissue cells, enabling the slower-growing osteoprogenitor cells to proliferate and form new bone in the bone defect site without interference. Therefore, such a membrane is the pivotal material for GBR technique. We accordingly added the following description in our revised manuscript.

“As the pivotal material for guided bone regeneration (GBR) technique to augment bone, GBR membranes applied between the gingiva and alveolar bone in the oral cavity can be considered as an ideal model for studying the biological applications of materials implanted at the soft-hard tissue interface⁵”

To help maintain the osteogenic space, the membrane may be supported with bone grafts (e.g., heterogenous particulate bone graft, heterogenous block bone graft, and autogenous block bone graft), implants, or tenting poles. Of course, with the development of 3D printing technology, designing an absorbable 3D-printed bone graft according to the shape of the bone defect to support the membrane will achieve a more predictable bone augmentation effect. It is worth emphasizing that our BC-g-PNCl/CS-HAP is not a simple stiff membrane, but rather a material that harmoniously combines stiffness and flexibility, ingrowth barrier and ingrowth guiding, as well as anti-bacteria and cell-activation. All the above characteristics of the membrane are beneficial for tissue integration, thus achieving good clinical outcomes.

In this study, we applied only GBR membranes without bone grafts in animal experiments. This approach was chosen to avoid potential interference in the analysis of osteogenesis results due to the radiopacity of bone grafts. The design is supported by several published research (*Adv. Mater.* 2016, 28, 4025; *Adv. Funct. Mater.* 2018, 28, 1703437; *Adv. Funct. Mater.* 2021, 31, 2101452).

3. The authors must explain better regarding in vivo results. Owing to the performance, the BC-g-PNCl/CS-HAP membrane is better than the collagen membrane, is only not the composition of the membranes but also the differences between thickness (not reported),

degradable mechanism of the products; in other words, there are more critical factors involved in these in vivo results that favored higher space maintenance and consequently a better outcome to bone neof ormation. Please, discuss these results more.

Thanks for your helpful suggestion. The superior capability of BC-g-PNCl/CS-HAP to repair bone defects can be mainly attributed to the long-lasting multiple protection mechanisms of BC-g-PNCl dense layer to successfully block the ingrowth of surrounding fast-growing connective tissues and bacteria, together with the pronounced osteogenic ability of CS-HAP loose layer to promote bone formation. Furthermore, due to a slow and partially degradable characteristic, BC-g-PNCl/CS-HAP exhibiting reasonable resistance to external forces in a wet state has only a low mass loss of 31% after immersion in PBS for up to 12 weeks (Fig. S18), ensuring long-lasting maintenance of osteogenic space. This could also be beneficial for BC-g-PNCl/CS-HAP to achieve better osteogenic effects. The thickness should not exert any significant influence, as the thickness of our membrane is basically the same as that of the collagen membrane in the control group.

Accordingly, we added the description of thickness differences, the results of mass loss experiments, as well as the discussion on the in vivo results in our revised manuscript.

“Based on the micromechanical interlock resulting from the penetration of CS-HAP solution into the dense layer network, the targeted product BC-g-PNCl/CS-HAP has a tightly bound bilayer structure (Fig. 2d), which can be maintained even under high-speed vibration (Movie 1). The thickness of BC-g-PNCl and CS-HAP layers is about 60 and 320 μm , respectively, making the overall thickness of BC-g-PNCl/CS-HAP similar to a commercial Bio-Gide[®] collagen membrane (CM, normally 300-500 μm)^{29,30}.”

“The superior capability of BC-g-PNCl/CS-HAP to repair bone defects can be mainly attributed to the long-lasting multiple protection mechanisms of BC-g-PNCl dense layer to successfully block the ingrowth of surrounding fast-growing connective tissues and bacteria, together with the pronounced osteogenic ability of CS-HAP loose layer to promote bone formation. Furthermore, due to a slow and partially degradable characteristic, BC-g-PNCl/CS-HAP exhibiting reasonable resistance to external forces

in a wet state has only a low mass loss of 31% after immersion in PBS for up to 12 weeks (Fig. S18), ensuring long-lasting maintenance of osteogenic space. This could also be beneficial for BC-g-PNCl/CS-HAP to achieve better osteogenic effects.”

Fig. S18 Degradation of BC-g-PNCl/CS-HAP membrane after incubation in PBS for 12 weeks (n = 3; error bars = s.d.).

4. The authors concluded, "…our BC-g-PNCl/CS-HAP breaks through the traditional concept that the GBR membrane can only shield soft tissue cells and proposes a new concept of all-in-one GBR membranes." However, N-halamine for coatings and grafting is not a new idea; there are patents and other articles that use this concept. Furthermore, bacterial cellulose + chitosan in the way composite membrane with or without hydroxyapatite has several articles in the literature. Then, the authors must better defend the concept of the new idea and process.

Thanks for your kind suggestions. Indeed, N-halamine, bacterial cellulose, chitosan and hydroxyapatite are commonly used materials, but it is difficult to obtain such a multifunctional GBR membrane by their simple composite. Therefore, hierarchical designs such as mechanical hierarchy, pore structure hierarchy, and anti-bacterial hierarchy play a leading role in achieving an all-in-one GBR membrane. To better defend the concept of our new idea, we provided the following detailed description in the last paragraph of the introduction section.

“Herein, based on the multiple adaptations including mechanics, pore structure, and anti-bacterial performance, we have created a spatiotemporally hierarchical GBR membrane, which can comprehensively solve the clinical issues throughout the whole

GBR processes (Fig. 1b). The dense layer of the membrane facing the gingiva shows Young's modulus comparable to that of the gingiva, so it has a sufficient space maintenance capacity and would not damage the gingival tissue or cause a stress shielding effect on the osteogenic area. The loose layer facing the alveolar bone is ultra-soft to conform to various morphologies of bone surfaces and seal the edge of the defect cavity. With the above asymmetric design, we integrate rigidity with flexibility to meet the desirable mechanical properties of the GBR membrane. In terms of the hierarchical porous structure, the small pores of the dense layer can continuously barrier both fibroblasts and bacteria, while the largely porous framework of the loose layer is suitable for the ingrowth of osteogenesis-associated cells. More importantly, through the asymmetric anti-bacterial function, the dense layer acts as a powerful defense with multiple mechanisms, including release-killing, contact-killing and anti-adhesion, to prevent bacterial invasion from the outside of the membrane; the loose layer moderately prevents bacteria leakage from other directions and creates a favorable osteogenic microenvironment. With a well-orchestrated combination of the hierarchical porous structure and multi-dimensional anti-bacterial function, the quintuple protections against bacteria are complementary and indispensable to resist bacterial invasion strongly and comprehensively. As a result, our all-in-one GBR membrane possesses full protection mechanisms and outstanding osteogenic promotion, making it a promising GBR material in clinical bone augmentation.”

Reviewer #2

In this study, the authors fabricated novel spatiotemporally hierarchical guided bone regeneration (GBR) membranes by rational bilayer integration of densely porous N-halamine functionalized bacterial cellulose nanonetwork facing the gingiva and loosely porous chitosan-hydroxyapatite composite micronetwork facing the alveolar bone. The BC-g-PNCl/CS-HAP membrane has a mechanically matched space maintenance capacity toward gingiva, continuously blocks fibroblasts, and prevents bacterial invasion with multiple mechanisms including release-killing, contact-killing, anti-adhesion, and nanopore-blocking. Even though the findings are interesting, the result and discussion part is not well focused. Some data should be further analyzed and even more data are needed to improve this manuscript. The following issues need to be clarified.

Thanks a lot for your great support and constructive suggestions. We carefully revised our manuscript and provide the following point-to-point response to address all your concerns.

1. The introduction is not interesting enough, and it should be rewritten to highlight the importance and novelty of this work.

Thanks for your constructive suggestion. We supplemented the following detailed description of the importance of this work in the introduction section of our revised manuscript.

“Nowadays, the most widely used GBR membranes in clinical practice are collagen membranes (CM), such as Geistlich Bio-Gide^{®8}, Zimmer CopiOs^{®9}, and ZH-Bio Heal-All^{®10}, which rely on the porous skeleton formed by stacking of collagen fibers to block soft tissue cell¹¹. Due to the variability of clinical cases and the microbial richness of the oral environment, the shortcomings of CM are increasingly magnified¹². Collagen could be swollen and softened by saliva and blood, resulting in limited space maintenance in practical applications^{13,14}. Moreover, infection is one of the main causes

of GBR failure, particularly in patients with periodontitis, maxillofacial defects and other poor microecological environment, as well as special patients with low immunity (e.g., diabetes). However, the commonly used CM lacks components or structures that resist bacterial invasion¹¹, let alone the quickly weakened barrier function caused by the rapid degradation of collagen¹⁵⁻¹⁷. Therefore, the existing GBR membranes are still far from the ideal soft-hard tissue interface biomaterial (Fig. 1a).”

2. Why choose chitosan to fabricate asymmetric porous bilayer BC-g-PNCl/CS-HAP membrane rather than biological materials? What is the function of it?

Thanks for your nice comment. The biological material is a material produced only by a biological system, such as collagen, spongin, silk, keratin, conchiolin, chitin, cellulose(tunicin), dentin enamel, bone (Marine Biological Materials of Invertebrate Origin, Springer International Publishing, 2019, 13, 4). Chitosan is prepared by deacetylation of chitin which is one of the most abundant polysaccharides in natural macromolecules and a typical component of crustaceans, mollusks, insect exoskeleton and fungal cell walls (*Adv. Mater.* 2023, 35, 2203325; *Int. J. Biol. Macromol.* 2020, 164, 4532). That is to say, chitosan is a typical biological material.

Chitosan (CS) is structurally similar to glycosaminoglycans, which are the major component of the bone extracellular matrix. Co-blending CS and hydroxyapatite (HAP) to form a loose layer (CS-HAP) can mimic the complementary combination of organic and inorganic components naturally present in bone. CS exhibits biocompatibility, low immunogenicity, and biodegradability. It can be degraded in vivo by lysozyme and other enzymes, and the degradation product is glucosamine, which serves as a nutrient for cellular metabolism. Also, CS has excellent processing performance and can be easily processed into loose structures with interconnected large pores, which are favorable for osteoblast infiltration and integration. Moreover, chitosan has mild antibacterial property from cationic amino groups and can inhibit the activity of bacteria that enter the defect cavity from the side of the membrane or infected areas inside the alveolar bone (*Adv. Funct. Mater.* 2018, 28, 1802818; *Nanoscale Horiz.* 2021, 6, 505).

Overall, CS is an ideal material for constructing the CS-HAP loose layer of BC-g-PNCl/CS-HAP membrane.

3. How about degradation rate of BC-g-PNCl/CS-HAP membrane? The degradation of BC-g-PNCl/CS-HAP should be provided.

Thanks for your kind suggestion. We studied the degradation of the BC-g-PNCl/CS-HAP membrane in PBS solution in vitro for up to 12 weeks. We added the following methods and the related new data in the revised supporting information.

“For the weight loss experiments, BC-g-PNCl/CS-HAP membranes were randomly divided into 8 groups (n = 3) after drying based on the total measurement time. Each membrane was weighed and recorded as m_0 (40 ± 10 mg), and was then placed into 5 mL of PBS (pH = 7.4) at 37 °C in a shaker box. The PBS solution was replaced every 3 days. At each predetermined time point, each membrane in one group was taken out from PBS, washed with distilled water for 3 times, dried for 24 h at 60 °C, and recorded as m_1 . The weight loss was calculated according to equation (1):

$$\text{Weight loss} = \frac{m_0 - m_1}{m_0} \times 100\% \quad (1)$$

It should be noted that we used Bio-Gide® collagen membrane (CM; Bio-Gide, Geistlich, Switzerland) as the representative of the commercial collagen membranes for measurements, considering it has been widely recognized as the gold clinical standard in GBR therapy.”

Fig. S18 Degradation of BC-g-PNCl/CS-HAP membrane after incubation in PBS for 12 weeks (n = 3; error bars = s.d.).

We also added the following description in the revised manuscript.

“Furthermore, due to a slow and partially degradable characteristic, BC-g-PNCl/CS-HAP exhibiting reasonable resistance to external forces in a wet state has only a low mass loss of 31% after immersion in PBS for up to 12 weeks (Fig. S18), ensuring long-lasting maintenance of osteogenic space. This could also be beneficial for BC-g-PNCl/CS-HAP to achieve better osteogenic effects.”

4. As one of the most attractive bactericidal agents, N-halamines have been utilized in fields of biomedicine except for GBR membranes. The release-killing rates of BC-g-PNCl against S. aureus and P. gingivitis in the medium are 73% and 69%, respectively (Fig. S9), suggesting that the first defense can be established by releasing anti-bacterial components before bacteria reach the membrane. What is the anti-bacterial components? Is it N-halamines? The study of anti-bacterial components release should be provided.

Thanks for your good comment. The anti-bacterial component of BC-g-PNCl/CS-HAP is positively charged chlorine atoms (Cl^+), which is dissociated from N-Cl bond firstly and then migrates to solution to exert releasing killing effect. N-halamine polymers are non-antibiotic organic polymeric agents, which possess anti-bacterial properties against a broad spectrum of bacteria. The active Cl^+ exerts the anti-bacterial effect in N-Cl polymers. On the one hand, Cl^+ can directly transfer from N-Cl to receptors of bacteria to achieve contact-killing activity; on the other hand, Cl^+ can dissociate from N-Cl and migrate to the surrounding solution to achieve the release-killing activity. Cl^+ can disrupt the integrity of the cell wall and cell membrane and inhibit the bacterial metabolic process. As a result, bacteria lose their functions and die. For N-Cl polymer brushes, the stable N-halamine kills bacteria mainly through the contact-killing mechanism, while the relatively unstable N-halamine dissociates the active Cl^+ and disperses into the solution, acting as an anti-bacterial agent through the releasing-killing mechanism (*Chem. Rev.* 2017, 117, 4806; *J. Am. Chem. Soc.* 2021, 143, 16777; *Ind. Crop. Prod.* 2022, 187, 115518). In this study, the results indicated

that the antimicrobial effect of the constructed N-halamine polymer coating was simultaneously achieved through a combination of contact and release killing mechanisms, providing dual anti-bacterial protection.

The dissociation rate of Cl^+ from N-halamine compounds is extremely slow. The type of N-halamine in the dense layer of our membrane is amide N-halamine. The stability of amide N-halamine lies between those of imine N-halamine and amine N-halamine, with a dissociation constant lower than 10^{-9} . A quantitative assessment had been conducted to evaluate the release of Cl^+ from N-halamine-modified materials containing 7007 ppm of active chlorine in an in vitro environment (*Biomater.* 2007, 28, 1597). The results revealed that over a duration of 6-95 hours, the equilibrium for N-Cl bond dissociation progressively reached a state of balance, resulting in a Cl^+ content of approximately 2.0 $\mu\text{g/ml}$ (2 ppm) in the final solution.

5. Why choose calvaria defect model for evaluation of bone regeneration in vivo, rather than bone defect model in oral cavity.

Thanks for your nice comment. From the perspective of osteogenesis pathways, the skull, mandible and maxilla belong to intramembranous ossification, which is characterized by the formation of bone directly from mesenchymal populations (*Nat. Rev. Mol. Cell Biol.* 2020, 21, 696). Therefore, many studies involving materials related to oral and craniofacial bone repair employ the calvaria defect model, in which critical-sized calvaria defect in rat is the most commonly used (*Adv. Mater.* 2016, 28, 4025; *Adv. Funct. Mater.* 2021, 31, 2101452; *Adv. Funct. Mater.* 2019, 29, 1900065). The rat calvaria defect model has many advantages, such as simple operation, easy standardization, high success rate, and low cost. Based on the above studies, we chose the calvaria defect model in rat to investigate the bone regeneration properties of BC-g-PNCl/CS-HAP in vivo before moving it to larger animals for potential translation to human applications.

6. Immunohistochemical analysis of in vivo osteogenesis (Col-1 or BMP-2) would be beneficial.

Thank you for your helpful suggestion. We have conducted an immunohistochemical analysis of *in vivo* osteogenesis. During the process of bone regeneration, the expression levels of osteogenic-related proteins change with the progression of bone formation. COL1 is an early differentiation marker that mediates osteoblasts' adhesion, proliferation, and differentiation (*Stem Cell Res. Ther.* 2017, 8, 65). BMP-2 promotes osteogenesis in the early stage and stimulates osteoclast differentiation in the later stage (*J. Bone Joint Surg.* 2003, 85,1243). Both Col1 and BMP-2 are primarily expressed in the early stages of osteogenesis. The *in vivo* osteogenesis experiment designed in this study set an osteogenic period of 8 weeks, forming a high degree of new bone mineralization. As shown in the following immunohistochemical staining images, the expression level of Col1 in the bone tissue of BC-g-PNCI/CS-HAP group was slightly higher than that in the CM group, but the expression of Col1 in all groups was relatively low. Therefore, we further tested the expression level of OPN, a late-stage marker of osteogenesis. OPN appears at the end of osteoblast differentiation and can bind to Ca²⁺ to regulate calcium ion homeostasis and bone mineralization (*Mat. Sci. Eng. C*, 2021, 124, 112087). As shown in the following images, regardless of the normal or bacterial model, the expression of OPN in the mineralized bone of the BC-g-PNCI/CS-HAP group was significantly higher than that in the CM group. In conclusion, the immunohistochemical results of *in vivo* osteogenesis demonstrated that BC-g-PNCI/CS-HAP had a protective and promoting effect on the bone regeneration process.

Fig. R1 COL1 and OCN immunohistochemical staining of calvarial decalcified sections after 8 weeks of implantation in both normal and bacterial models (Scale bars = 100 μm).

7. Discussion is unfocused and too easy; authors should include the novelty of the present study compared to other studies in the discussion part as there are many studies on GBR membranes.

Thanks for the kind suggestion. We have described the limit of current studies in the introduction and discussion sections of the main text as follows:

“Efforts have been made to address the aforementioned issues, but most efforts are largely limited to addressing individual issues rather than considering them comprehensively. For example, titanium mesh is used to improve space maintenance, but it’s too rigid, easily causing gingiva to dehiscence and not attach to the bone surface to form a seal¹⁸. Some GBR membranes have been endowed with anti-bacterial properties, but they neglect the complexity and spatiotemporal characteristics of bacterial infection and then fail to effectively resist bacterial invasion at all stages (including adhesion, colonization, and penetration) and in all directions¹⁹⁻²¹. Therefore, how to delicately couple the seemingly opposite mechanical, structural, and biological properties in a GBR membrane to provide full protection during bone regeneration is a vital scientific and clinical question.”

“The oral cavity is one of the most complex regions in the body, possessing multiple contradictory factors such as the combination of soft and hard tissues, adjacency to internal and external environments, and richness in bacteria and stem cells⁶⁰⁻⁶². However, the existing GBR membranes with a single structure or function cannot cope with this paradoxical condition.”

According to your kind advice, we also supplemented the following description of the novelty of the present study compared to other studies in the discussion section.

“However, the existing GBR membranes with a single structure or function cannot cope with this paradoxical condition. **Especially, they could fail to achieve satisfactory integration in terms of antibacterial ability, mechanical property, and porosity^{5,7,11,20}**. In this study, we demonstrate a novel class of spatiotemporally hierarchical GBR membranes by comprehensively integrating contradictory structures and properties.”

Reviewer #3

In this manuscript, the authors have successfully developed a novel spatiotemporally hierarchical guided bone regeneration (GBR) membrane. Different from the existing GBR membranes, this new membrane integrates densely porous N-halamine functionalized bacterial cellulose nanonetwork facing the gingiva and loosely porous chitosan-hydroxyapatite composite micro-network facing the alveolar bone. Therefore, the new GBR membrane asymmetrically combines stiffness and flexibility, ingrowth barrier and ingrowth guiding, as well as anti-bacterial and cell-activation, which have been verified by comprehensive and reasonable in vitro and in vivo experiments. The interesting results move forward the development of GBR membrane and also provide a typical paradigm for the design of complex biomaterials. Overall, this research has a high novelty and clinical value, and the manuscript is well-written. So I suggest a minor revision after addressing the following concerns:

We really appreciate your encouraging comments and constructive suggestions for improving our work. We have carefully addressed all your concerns in our revised manuscript and provided the following point-to-point response.

1. The authors should provide a more detailed description of the shortcomings of existing GBR membranes in clinical applications to highlight the clinical value of their study in the introduction section.

Thanks for your constructive suggestion. We supplemented the following detailed description of the importance of this work in the introduction section of our revised manuscript.

“Nowadays, the most widely used GBR membranes in clinical practice are collagen membranes (CM), such as Geistlich Bio-Gide^{®8}, Zimmer CopiOs^{®9}, and ZH-Bio Heal-All^{®10}, which rely on the porous skeleton formed by stacking of collagen fibers to block soft tissue cell¹¹. Due to the variability of clinical cases and the microbial richness of the oral environment, the shortcomings of CM are increasingly magnified¹². Collagen

could be swollen and softened by saliva and blood, resulting in limited space maintenance in practical applications^{13,14}. Moreover, infection is one of the main causes of GBR failure, particularly in patients with periodontitis, maxillofacial defects and other poor microecological environment, as well as special patients with low immunity (e.g., diabetes). However, the commonly used CM lacks components or structures that resist bacterial invasion¹¹, let alone the quickly weakened barrier function caused by the rapid degradation of collagen¹⁵⁻¹⁷. Therefore, the existing GBR membranes are still far from the ideal soft-hard tissue interface biomaterial (Fig. 1a).”

2. *Since graft density is an important chemical structural parameter, the authors should provide the content of PAM in BC-g-PAM.*

Thanks for your kind suggestion. We carried out the elemental analysis experiments of samples before and after grafting (i.e., BH-KH570 and BC-g-PAM). According to the results of elemental analysis in Table R1, the weight percentage of grafted PAM side-chain for BC-g-PAM can be calculated to be 66 wt%, indicating a high graft density.

Table R1 Elemental analysis of BC-KH570 modified and BC-g-PAM

Sample	Carbon [wt%]	Oxygen [wt%]	Nitrogen [wt%]
BC-KH570	43.39	24.29	0
BC-g-PAM	44.58	32.77	13.01

3. *For the CS-HAP loose layer, the authors only characterized the presence of calcium and phosphorus in HAP by elemental mapping. They should further provide the presence of chitosan in the CS-HAP loose layer.*

Thanks for your nice suggestion. We have complemented the corresponding experiment. The revised descriptions and the related new data are as follows:

“As shown in elemental mapping in Fig. S1, the chlorine elements from N-halamine polymers are homogenously distributed in BC-g-PNCl layer, while the phosphate,

calcium, carbon and nitrogen elements from HAP and CS are evenly distributed on CS-HAP layer.”

Fig. S1 Elemental distributions of BC-g-PNCl/CS-HAP membrane. **a** Elemental mapping of Cl (**a**) in the BC-g-PNCl layer (scale bar = 1 μm), showing Cl element was evenly grafted in BC-g-PNCl layer. **b-e** Elemental mapping of P (**b**), Ca (**c**), C (**d**) and N (**e**) in the CS-HAP layer (scale bars = 10 μm), showing HAP particles and CS were well composited in CS-HAP layer.

4. How were the Young's moduli in Fig. 3g obtained? The details of testing and analysis should be added.

Thanks for your good suggestion. As stated in the revised supporting information, the Young's modulus of BC-g-PNCl/CS-HAP and collagen membrane in a wet state was measured by AFM in the peak force quantitative nanomechanics mode and analyzed by the Derjaguin-Muller-Toporov (DFT) model. To closely simulate the practical wet environment of the samples as much as possible, before Young's modulus tests, the samples were sterilized with ethylene oxide and soaked in saline for at least 1h, and the excess saline on the membrane surface was lightly wiped off. The Young's modulus on the dense layer surface of BC-g-PNCl/CS-HAP was calculated from the AFM Young's modulus mapping. However, due to the low Young's modulus on the loose layer surface of BC-g-PNCl/CS-HAP, it was quite difficult to form images of the AFM Young's modulus mapping on the wet surface. For this reason, the Young's modulus of loose

layer surface in wet state was obtained by calculating the average value after measuring multiple force curves with the same peak force quantitative nanomechanics mode and analyzed by the same DMT model (*Recent Pat. Nanotech.* 2011, 5, 27).

5. For the mechanical properties, the authors only measured Young ' s moduli and performed bending and twisting tests. It is recommended to detect the tensile properties further.

Thanks for your helpful suggestion. We have complemented the corresponding experiment. The revised descriptions and the related new data are as follows:

“The wet BC-g-PNCl/CS-HAP membrane exhibits higher tensile strength and smaller elongation at break compared to CM (Fig. S4). Different from CM which is easy to wrinkle and difficult to handle⁴⁵, BC-g-PNCl/CS-HAP can remain in a spreading state due to its appropriate rigidity and is also flexible enough to retain its original shape even after repeated bending and twisting (Fig. 3j).”

Fig. S4 Tensile stress-strain curves of BC-g-PNCl/CS-HAP membrane and CM in the wet state.

6. For the biocompatibility tests, the authors only conducted *in vitro* studies. How is the biocompatibility of the GBR membrane *in vivo*? Why did they not carry out the *in vivo* biocompatibility experiments?

Thanks for your nice suggestion. In this study, we investigated the effect of BC-g-PNCI/CS-HAP on bone tissue regeneration by establishing a long-term rat calvarial bone defect model. The results of micro-CT analyses showed that the BC-g-PNCI/CS-HAP membrane had a promotive effect on bone regeneration in the bone defect area after implantation for 8 weeks. Simultaneous histological staining showed no inflammatory infiltration existed in BC-g-PNCI/CS-HAP. That is to say, the current in vivo results have clearly demonstrated the good biocompatibility of our membrane.

7. The authors pointed out that the loose layer was favorable for osteoblast proliferation. What about the effect of dense surfaces on fibroblast proliferation?

Thanks for your helpful suggestion. We have complemented the corresponding experiment and accordingly added some words into the revised manuscript as follows:

“The distribution of BMSCs cultured on the loose layer and L929 fibroblasts cultured on the dense layer for 3 and 7 days were also observed.”

“The results of CCK-8 assay (Fig. 4a), cytoskeleton arrangement (Fig. S5 and S6), bone mesenchymal stem cells (BMSCs) ingrowth (Fig. S7) and L929s distribution (Fig. S8) indicate good cytocompatibility of BC-g-PNCI/CS-HAP.”

Fig. S8 Proliferation and distribution of L929s. Top-view fluorescence images exhibiting the proliferation area of L929s cultured on dense layer of CM (a) and BC-g-PNCI layer of BC-g-PNCI/CS-HAP (b) at the 3rd and 7th days, respectively (blue for cell nucleus, scale bars = 200 μ m).

8. The authors carried out anti-biofilm experiments to compare the adhesion of bacteria on dense surfaces. It is recommended to more visually display the effect of dense layers on biofilm thickness, similar to the 3D cross-section display of cell penetration in barrier experiments.

Thanks for your constructive suggestion. We have complemented the corresponding experiment and added the corresponding words as follows:

“Such good anti-adhesion and anti-biofilm capacities of various components of BC-g-PNCI are also well supported by SEM images (Fig. S16) and three-dimensional fluorescent images (Fig. S16).”

Fig. S16 Anti-biofilm capacities. Three-dimensional fluorescent images exhibiting the bacterial adhesion situation of *S. aureus* on the surfaces of CM (a), BC (b), BC-g-PAM (c), and BC-g-PNCI (d). The depth of biofilm on the surfaces of the CM dense layer, BC, BC-g-PAM, and BC-g-PNCI is gradually reduced (green for live bacteria, red for dead bacteria, scale bars = 200 μm).

9. How is the degradation property of the GBR membrane?

Thanks for your kind suggestion. We studied the degradation of the BC-g-PNCI/CS-HAP membrane in PBS solution in vitro for up to 12 weeks, and added the following sentences in the revised supporting information:

“Before Young’s modulus, tensile tests, loading, bending, and twisting tests, the samples were sterilized with ethylene oxide and soaked in saline for at least 1h, and the excess saline on the membrane surface was lightly wiped off. For the weight loss

experiments, BC-g-PNCl/CS-HAP membranes were randomly divided into 8 groups (n = 3) after drying based on the total measurement time. Each membrane was weighed and recorded as m_0 (40 ± 10 mg), and was then placed into 5 mL of PBS (pH = 7.4) at 37 °C in a shaker box. The PBS solution was replaced every 3 days. At each predetermined time point, each membrane in one group was taken out from PBS, washed with distilled water for 3 times, dried for 24 h at 60 °C, and recorded as m_1 . The weight loss was calculated according to equation (1):

$$\text{Weight loss} = \frac{m_0 - m_1}{m_0} \times 100\% \quad (1)$$

It should be noted that we used Bio-Gide[®] collagen membrane (CM; Bio-Gide, Geistlich, Switzerland) as the representative of the commercial collagen membranes for measurements, considering it has been widely recognized as the gold clinical standard in GBR therapy.”

Fig. S18 Degradation of BC-g-PNCl/CS-HAP membrane after incubation in PBS for 12 weeks (n = 3; error bars = s.d.).

We also added the following description in the revised main text.

“Furthermore, due to a slow and partially degradable characteristic, BC-g-PNCl/CS-HAP exhibiting reasonable resistance to external forces in a wet state has only a low mass loss of 31% after immersion in PBS for up to 12 weeks (Fig. S18), ensuring long-lasting maintenance of osteogenic space. This could also be beneficial for BC-g-PNCl/CS-HAP to achieve better osteogenic effects.”

10. What is the bactericidal mechanism of the released antibacterial agent? Is it harmful to cells?

Thanks for your nice comment. N-halamine polymers are non-antibiotic organic polymeric agents, which possess anti-bacterial properties against a broad spectrum of bacteria. The active Cl^+ exerts the anti-bacterial effect in N-Cl polymers. On the one hand, Cl^+ can directly transfer from N-Cl to receptors of bacteria to achieve contact-killing activity; on the other hand, Cl^+ can dissociate from N-Cl and migrate to the surrounding solution to achieve the release-killing activity. Cl^+ can disrupt the integrity of the cell wall and cell membrane and inhibit the bacterial metabolic process. As a result, bacteria lose their functions and die. For N-Cl polymer brushes, the stable N-halamine kills bacteria mainly through the contact-killing mechanism, while the relatively unstable N-halamine dissociates the active Cl^+ and disperses into the solution, acting as an anti-bacterial agent through the releasing-killing mechanism (*Chem. Rev.* 2017, 117, 4806; *J. Am. Chem. Soc.* 2021, 143, 16777; *Ind. Crop. Prod.* 2022, 187, 115518). In this study, the results indicated that the antimicrobial effect of the constructed N-halamine polymer coating was simultaneously achieved through a combination of contact and release killing mechanisms, providing dual anti-bacterial protection.

Although N-halamines have powerful antibacterial activity toward a broad spectrum of bacteria, most of them exhibit excellent biocompatibility to mammalian cells. Nowadays, N-halamines have been widely utilized in fields of biomedical materials, such as wound dressing, blood disinfection material, and dental implant (*Compos. Part B-Eng.* 2021, 227, 109390; *Chem. Eng. J.* 2021, 415, 128888; *Nat. Commun.* 2021, 12, 3303). In the previous study, our team successfully constructed a porous N-halamine polymeric coating on the titanium surface. We found that this coating has no adverse effects not only on the proliferation and differentiation of osteoblasts in vitro, but also on osseointegration in rabbits (*Nat. Commun.* 2021, 12, 3303). In this study, we have also confirmed that N-halamine has no adverse effects on cell activity and morphology by CCK-8 and immunofluorescence assays.

11. The authors mentioned that the prepared GBR membrane was spatiotemporally hierarchical. Relevant experiments or discussions are required to explain this property.

Thanks for your helpful suggestion. The spatially hierarchical design is manifested as the asymmetric design in terms of pore structure, mechanics, and antibacterial performance between the dense layer facing soft tissue and the loose layer facing hard tissue, as shown in Fig. 2e-f, Fig. 3g, Fig. 5 and Fig. S18. The temporally hierarchical design is characterized by the in-situ conversion of anti-bacterial N-halamine polymer into PAM with anti-adhesive property, matching the functional requirement of membrane at different periods in the same space, as shown in Fig. 5c and Fig. S14-17. The revised discussions in the main text are as follows.

“In this study, we demonstrate a novel class of spatiotemporally hierarchical GBR membranes by comprehensively integrating contradictory structures and properties. The spatiotemporally hierarchical design is manifested as the asymmetric design in terms of mechanics, pore structure, and antibacterial performance.”

Reviewers' Comments:

Reviewer #2:

Remarks to the Author:

In this study, the authors fabricated novel spatiotemporally hierarchical guided bone regeneration (GBR) membranes by rational bilayer integration of densely porous N-halamine functionalized bacterial cellulose nanonetwork facing the gingiva and loosely porous chitosan-hydroxyapatite composite micronetwork facing the alveolar bone. The BC-g-PNCl/CS-HAP membrane has a mechanically matched space maintenance capacity toward gingiva, continuously blocks fibroblasts, and prevents bacterial invasion with multiple mechanisms including release-killing, contact-killing, anti-adhesion, and nanopore-blocking. The work is original, and the findings are interesting. The methodology is sound and the methods are detail. The results supported the conclusions and claims. Their work provides an innovative direction for the development of novel GBR membranes, which is of significance to guided bone regeneration.

Reviewer #3:

Remarks to the Author:

[Note from the Editor: Reviewer #3 was asked to review the response given to the original Reviewer #1.].

All the questions raised by the reviewers have been fully answered, and the referee suggests an acceptance without further changes.

Response to Referees' Comments

Reviewer #2

In this study, the authors fabricated novel spatiotemporally hierarchical guided bone regeneration (GBR) membranes by rational bilayer integration of densely porous N-halamine functionalized bacterial cellulose nanonetwork facing the gingiva and loosely porous chitosan-hydroxyapatite composite micronetwork facing the alveolar bone. The BC-g-PNCl/CS-HAP membrane has a mechanically matched space maintenance capacity toward gingiva, continuously blocks fibroblasts, and prevents bacterial invasion with multiple mechanisms including release-killing, contact-killing, anti-adhesion, and nanopore-blocking. The work is original, and the findings are interesting. The methodology is sound and the methods are detail. The results supported the conclusions and claims. Their work provides an innovative direction for the development of novel GBR membranes, which is of significance to guided bone regeneration.

Thanks a lot for taking valuable time to review our manuscript. We really appreciate your constructive suggestions for enriching our work. We have carefully revised and formatted our manuscript.

Reviewer #1& Reviewer #3

[Note from the Editor: Reviewer #3 was asked to review the response given to the original Reviewer #1.]

All the questions raised by the reviewers have been fully answered, and the referee suggests an acceptance without further changes.

We really appreciate your encouraging comments and constructive suggestions for improving our work. We have carefully revised and formatted our manuscript.